# A potential cephalopod from the early Cambrian of eastern Newfoundland, Canada

Anne Hildenbrand [1,3✉], Gregor Austermann [1,3✉], Dirk Fuchs[2], Peter Bengtson[1] & Wolfgang Stinnesbeck[1]

Although an early Cambrian origin of cephalopods has been suggested by molecular studies, no unequivocal fossil evidence has yet been presented. Septate shells collected from shallow-marine limestone of the lower Cambrian (upper Terreneuvian, c. 522 Ma) Bonavista Formation of southeastern Newfoundland, Canada, are here interpreted as straight, elongate conical cephalopod phragmocones. The material documented here may push the origin of cephalopods back in time by about 30 Ma to an unexpected early stage of the Cambrian biotic radiation of metazoans, i.e. before the first occurrence of euarthropods.

[1] Institute of Earth Sciences, Heidelberg University, Im Neuenheimer Feld, Heidelberg, Germany. [2] Bavarian State Collection for Paleontology and Geology, Richard-Wagner-Straße, Munich, Germany. [3] These authors contributed equally: Anne Hildenbrand, Gregor Austermann ✉email: anne.hildenbrand@geow. uni-heidelberg.de; gregor.austermann@geow.uni-heidelberg.de

Cephalopods attract wide interest owing to their cognitive abilities, sophisticated adaptations, and their ecological competition with marine vertebrates. They are the most highly organised and complex class of molluscs, with a wealth of extant marine taxa spanning from shallow to abyssal water depths[1,2]. The ancestors of squids, cuttlefish, and octopuses originally possessed a chambered shell as the pearly *Nautilus* demonstrates. The successful evolutionary history of cephalopods started in the Cambrian. Even though these earliest precursors were probably benthic[1], their siphuncle allowed the shells to become gas-filled. This newly gained buoyancy initially kept the shells of crawling animals in an upright position and allowed more progressive post-Cambrian taxa to occupy the water column[2–4].

Most authors consider members of the order Plectronocerida to represent the oldest undoubted cephalopods[2,5,6], with the mid late Cambrian *Plectronoceras cambria*[7] as the oldest representative[8,9]. In members of late Cambrian Ellesmerocerida, the shell is divided into chambers by concave-shaped septa, and the siphuncle is relatively wide and located marginally along the ventral side of the phragmocone[10].

Here we present new material from the Avalon Peninsula of southeastern Newfoundland, Canada, which shows morphological shell features that characterise early cephalopods. The specimens (Figs. 1–4; Supplementary Figs. 2 and 3) were collected from a coquina deposit at Bacon Cove on the southwestern side of Conception Bay (Figs. 5 and 6). The mudstone-dominated succession is assigned to the uppermost Bonavista Formation[11,12] or the Cuslett Formation of the Bonavista Group[13,14], and is early Cambrian (Terreneuvian) in age (c. 522 Ma, *Camenella baltica* Zone; Fig. 7)[13–15]. Our material thus predates the oldest trilobites in Newfoundland (Fig. 7), and presumably worldwide.

**Regional and local geological setting**. The Avalon Peninsula in eastern Canada is widely known for its extensive Proterozoic–lower Palaeozoic sedimentary sequence, with Cambrian strata unconformably overlying a Cryogenian–Ediacaran succession deposited in a prograding back-arc basin[11,15–17]. The Precambrian succession comprises (1) basin-floor and turbidite deposits (Conception Group), (2) slope to delta transitional deposits (St. John's Group), and (3) shallow-marine and alluvial deposits (Signal Hill Group)[17–20]. The Precambrian–Cambrian transition is conformable and well exposed at the Global Stratotype Section and Point (GSSP) at Fortune Head on the southern Burin Peninsula, west of the Avalon Peninsula[21,22]. The Cambrian of the Avalon Peninsula is represented by subaerial to subtidal (Random Formation, lower Adeyton Group, and upper Harcourt Group) and open-shelf deposits (upper Adeyton to lower Harcourt groups)[14,23,24]. The rocks are mainly siliciclastic and consist of conglomerate, sandstone, siltstone and mudstone, whereas limestone is rare and restricted to a c. 4–8 m thick reef-type unit in the Smith Point Formation[11,15,25].

A hiatus of about 50 million years between the Drook and Bonavista formations at Bacon Cove is attributed to the syndepositional Avalonian Orogeny, combined with a terminal Ediacaran regression and subsequent Terreneuvian NW–SE transgression[13,25]. The Drook Formation at Bacon Cove comprises c. 3-m-thick grey-green mudstones and siltstones interpreted as remains of a turbidite succession[26]. A c. 5–15-mm-thick red basal conglomerate marks the onlap of the Cambrian Bonavista Formation (Figs. 5b and 7). Thin conglomerate and limestone layers intercalated with coarse red sandstone, siltstone and mudstone with in situ stromatolites[12] and microbialites characterise the lowermost c. 2.2 m of the Bonavista Formation at Bacon Cove. Upsection, a c. 4.2-m-thick red limestone is erosively interlayered by a c. 10-mm-thick coarse sandstone bed at c. 3.3 m and unconformably underlies the c. 4-m-thick biostromal limestone of the Smith Point Formation. The Bonavista Formation is interpreted as a shallow-marine onlap deposit resulting from a NW–SE transgression[15].

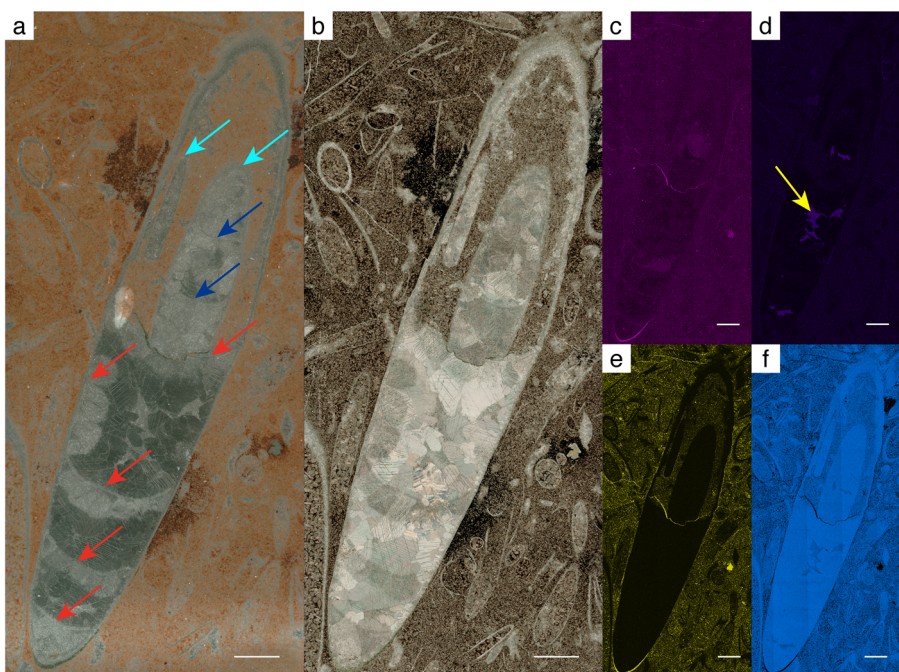

**Fig. 1 Thin section and SEM-EDS images of specimen no. NFM F-2774. a** Front light. Arrows: Green = Canals; Red = Septa; Blue = Septa in a possible other specimen. **b** Under crossed nicols. **c–f** Colour-coded SEM-EDS element mappings; a bright colour indicates high amounts of the respective element. The shell and parts of the siphuncle are composed of calcite. **c** Magnesium. The apical shell portion and the septa are enriched in Mg. **d** Manganese. Mn-deposits are identified in phragmocone chambers (yellow arrow). The apical shell portion is partly enriched in Mn. **e** Aluminium. The interior of the apical phragmocone chamber is enriched in Al. **f** Calcium. The apical shell portion is depleted in Ca. Scale bar is 1 mm.

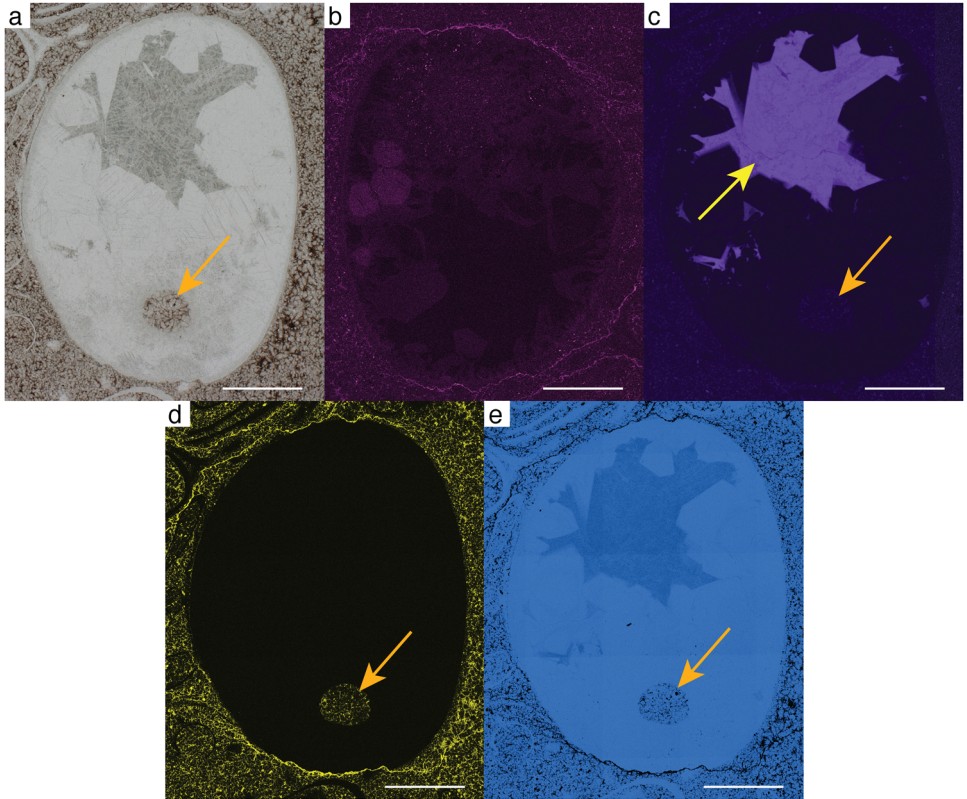

**Fig. 2 Thin section and SEM-EDS images of specimen no. NFM F-2776. a** Cross section of phragmocone showing the position of the siphuncle (orange arrow). Note that calcite spar cement filled the shell interior indicating that it formed a closed system during the earliest diagenetic stage. Only the siphuncle was filled with mud, thus providing strong evidence for its connection with the body chamber. **b**–**e** Colour-coded SEM-EDS element mappings of NFM F-2776. **b** Magnesium. The chamber and parts of the shell are partly enriched in Mg. **c** Manganese. Only the inner part is enriched in Mn (yellow arrow), whereas the siphuncle (orange arrow) is Mn-depleted. **d** Aluminium. The siphuncle (orange arrow) and the host rock are enriched in Al, whereas the phragmocone is Al-depleted. **e** Calcium. The phragmocone is enriched in Ca. Scale bar is 1 mm.

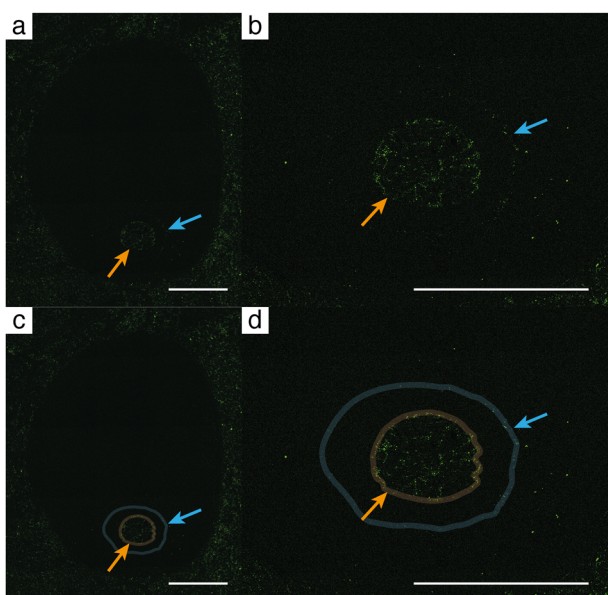

**Fig. 3 SEM-EDX image showing the distribution of phosphor in NFM F-2776. a**, **c** Phosphor enrichment reveals the position of the inner (orange arrow) and outer wall (blue arrow) of the connecting ring. **b**, **d** Detailed view of **a** and **c**. Scale bar is 1 mm.

## Results and discussion

**Locality**. The fossil site at Bacon Cove is situated in a small cove (Lower Cove) on the southwestern side of Conception Bay (47°29′05.3″N 53°09′58.1″W), southeastern Newfoundland, Canada, on lands belonging to the Canadian Crown (Figs. 5 and 6). The outcrop spans c. 200 m of the NNW–SSE-aligned coast.

**Material**. Specimen no. NFM F-2774 (Fig. 1; Supplementary Figs 2 and 4a, c) is represented by a calcareous shell with a visible height of c. 14 mm and a maximum width of c. 3 mm. The outer and inner shell walls are smooth, without ornamentation. Five slightly concave septa are visible in the apical portion of the shell (Fig. 1a). The chambered part is c. 7 mm high. Septal distances vary from c. 0.9 to 1.8 mm (first to last spacing starting from the apical end: 0.86, 1.34, 1.77, 1.53 and 1.75 mm), thereby suggesting a ratio of chamber height to diameter of c. 0.5–0.6. The apertural portion (c. 7 mm) of the shell is hollow and is interpreted as the body chamber.

A narrow canal, c. 0.4 mm in diameter, is visible in a submarginal position in the body chamber, following its longitudinal axis. The canal is filled with calcite cement. A second canal with a visible height of c. 4.5 mm occurs in a central position of the body chamber. It also follows the longitudinal axis but is substantially wider, c. 1.5 mm. The section plane of this specimen does not cut the siphuncle, implicating a sagittal rather than longitudinal section. It is possible that the apertural and

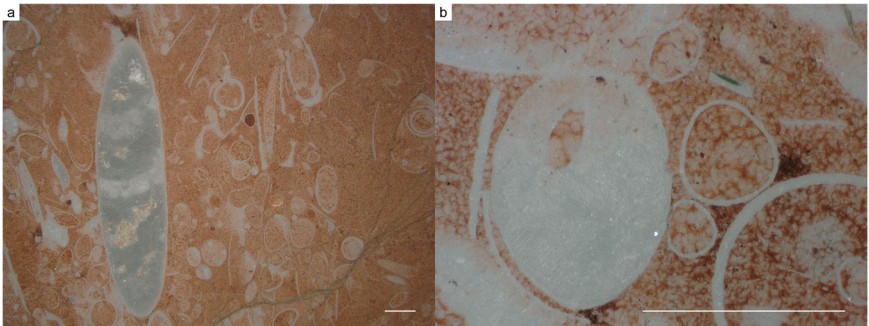

**Fig. 4 Thin section images of specimens NFM F-2775 and F-2777. a** No. NFM F-2775. Sagittal section. **b** No. NFM F-2777. Cross section. Scale bar is 1 mm.

apical ends of the shell are missing in the specimen or occur outside the documented field.

Specimen no. NFM F-2775 (Fig. 4a; Supplementary Fig. 4a, c) is a fragment of an ellipsoidal sagittal section, 9.6 mm high and 2.3 mm in maximum width. The interior is filled with calcite cement and the outer and inner shell surfaces are smooth. Two slightly concave septa cross the shell perpendicularly to the longitudinal shell axis. The distance between the two septa is 2.2 mm.

Specimen no. NFM F-2776 (Figs. 2 and 3; Supplementary Figs. 3 and 4b, c) represents a cross section of a laterally slightly compressed shell, 4.6 mm long and 3.6 mm wide, with a ratio of c. 1.3. The shell is filled with calcite cement, except for a small, submarginal, circular spot filled with Mg-K-Al-rich clay minerals and a calcium carbonate matrix; these potentially represent remains of a siphuncle. The shell wall is c. 0.08 mm thick. The possible siphuncle is 0.5 mm wide, 0.6 mm long, and located 0.6 mm from the shell wall.

Specimen no. NFM F-2777 (Fig. 4b; Supplementary Fig. 4b, c) is a cross section with a possible submarginal siphuncle. The oval-shaped shell fragment is 1.2 mm long and 0.9 mm wide, with a ratio of c. 1.3. The shell wall is c. 0.03 mm thick. The shell is filled with calcite cement, except for the siphuncle which is filled with matrix. The possible siphuncle has a visible length of 0.4 mm, a width of 0.2 mm, and is located 0.2 mm from the shell wall.

Our material is associated with a diverse assemblage of small shelly fossils (SSFs) containing mainly chancelloriids (Supplementary Fig. 1a, b), *Anabarites* Missarzhevsky, 1969[27] (Supplementary Fig. 1c) and ornamented orthoconic fossils (Supplementary Fig. 1d).

**Interpretation**. We interpret the septate apical (posterior) part of specimen no. NFM F-2776 as the phragmocone and the unsegmented apertural (anterior) part as the body chamber, arguably of a cephalopod. Specimens nos. NFM F-2774 and NFM F-2775 are sagittal sections of a longiconic shell. The presence of a phragmocone in these two specimens and pierced chambers in specimens nos. NFM F-2776 and NFM F-2777 support our interpretation of a cephalopod origin of our material. As there is no connection of the two ellipsoid shell remains in the body chamber of NFM F-2774, they are interpreted as sagittal sections of other, possibly conspecific specimens. They were probably washed into the empty body chamber during transport and/or deposition. This interpretation is also supported by the absence of these shells in the other specimens. We interpret the two phosphorous rings bounding the siphuncle in NFM F-2776 as remains of the inner and outer walls of a connecting ring (Fig. 3). It is unlikely to produce such a structure only with cone-in-cone, telescoping[28], or similar depositional or diagenetic effects. In late

Cambrian cephalopods the connecting ring is composed of calcitic material[6,10,29]. The outer layer is usually spherulitic-prismatic and the inner layer calcified-perforate[29]. Phosphorous connecting rings are presently known only from the Ordovician *Bactroceras*[30–32]. Detailed structures of the connecting ring as described by Mutvei[29] are lost, as the connecting ring is preserved as phosphorous remains visible in SEM-EDX maps and affected by diagenetic overprint. As our specimens lack sufficient details, a specific taxonomic determination is impossible, a situation similar to that of material described by Landing and Kröger[33]. The presence of a siphuncle, septal necks and a connecting ring are commonly regarded as key characteristics for the distinction of early fossil cephalopods from other septate or chambered organisms[1]. However, some authors have also assigned fossil shell material lacking these features to cephalopods[33,34].

The present material experienced slight deformation of the shells. The compression of septa identified here and in the outer shell of NFM F-2774 can thus be explained by the taphonomic preservation of the material. It is possible that the somewhat odd oval morphologies of NFM F-2776 and NFM F-2777 are also the effect of slight taphonomic deformation.

To date, the middle late Cambrian *Plectronoceras cambria*[7] is widely accepted as the oldest known cephalopod[1,35,36]. The taxon was assigned to the order Plectronocerida, a basal branch of Cephalopoda characterised by a ventral siphuncle. In contrast to our specimens, *P. cambria* has a short, endogastrically curved phragmocone with short, straight septal necks and a siphuncular bulb[37,38]. However, the few specimens of *P. cambria* documented to date are all fragmentary. Thin sections of *P. cambria* differ from our material by showing more densely spaced septa. Septate mollusc shells existed since early Cambrian times (c. 530 Ma[39]), such as *Tannuella*[27], which was originally regarded as an ancestral cephalopod but is now assigned to the monoplacophoran family Helcionellidae[1,40]. The shells are breviconic, straight, up to 40 mm long, and septate in the apical part[40]. There is no siphuncle. *Tannuella* may have been an ancestor of the late Cambrian *Knightoconus*[27,38], which is now also assigned to the Monoplacophora on account of its absence of a siphuncle[1,40].

The enigmatic early Cambrian *Salterella*[41] is known from several locations worldwide, including western Newfoundland and Labrador[42]. These small fossils have a conical calcareous shell filled with stratified laminae and a hollow central canal crossing these laminae to the apical end[42]. In contrast to our material, the septa consist of agglutinated grains and true chambers are absent. On the basis of these characters, *Salterella* has been included in the extinct phylum Agmata[43], along with the early Cambrian *Volborthella*[44] and the middle Cambrian *Ellisell yochelsoni*[45]. These taxa are characterised by their inclined agglutinated laminae within a calcareous cone.

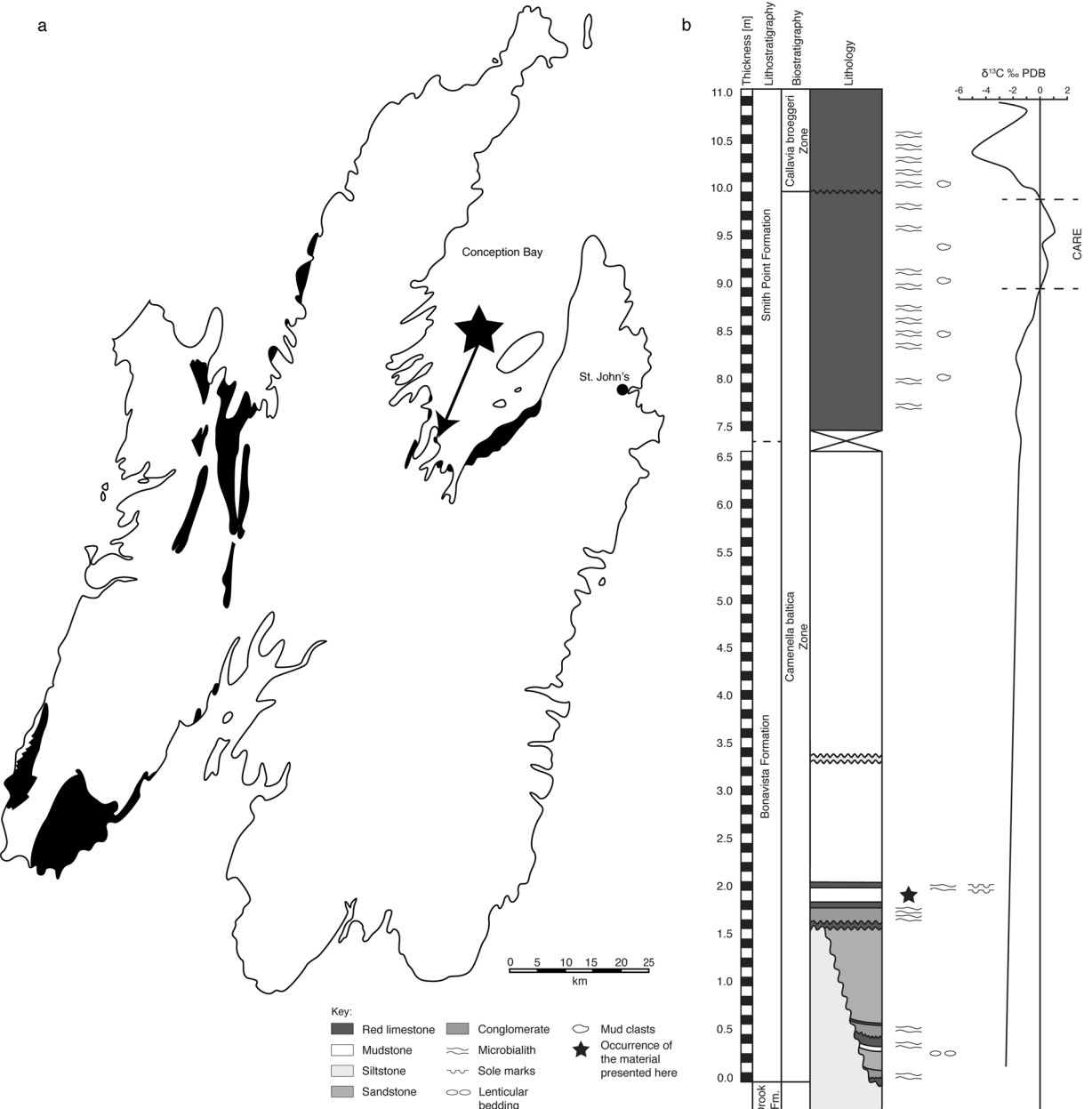

**Fig. 5 Regional and local context of our material. a** Lower Cambrian exposures (black) on the Avalon Peninsula, Newfoundland; data from Hutchinson[11], Landing and Benus[13], Landing et al.[15] and King[18]. Locality Bacon Cove marked by a star. **b** Lower Cambrian section at Bacon Cove; $\partial^{13}$C data from King[18]. The positive $\partial^{13}$C excursion[83] c. 7.0 m above the level with the here described specimens marks the Cambrian Arthropod Radiation Excursion (CARE) in the Smith Point Formation. The level of the oldest occurrence of trilobites on the peninsula lies above an intra-Smith Point Formation s.str.[11] unconformity, which is the base of the Brigus Formation s.str.[13,14]. This unconformity marks the end of the CARE. It corresponds to the boundary between the *Camenella baltica* (sub-trilobitic) and *Callavia broeggeri* zones and is thus a significant biostratigraphic marker[13,14,84]. The Bonavista Formation s.str.[11], as also the entire Bonavista Group s.str.[13,14], corresponds to the sub-trilobitic lower Cambrian, whereas the overlying Brigus Formation s.str.[13,14] represents the first trilobite-bearing lower Cambrian unit in Newfoundland[13,14,85,86]. PDB Pee Dee Belemnite, Fm Formation.

The superficial morphological similarity of our material with hyoliths is remarkable. *Allatheca degeeri*[46], a chambered orthothecid hyolith, was described from the lower Cambrian Cuslett Formation at Keels in northeastern Newfoundland[25] and is approximately coeval with our specimens. It is a large (120 mm long) species with a straight, elongate, conical shell. For a long time, hyoliths have been either regarded as an own phylum[47,48] or assigned to cephalopods[49,50]. An affinity with the lophophorates has recently been proposed, based on the discovery of rare soft-body features interpreted as lophophores in *Haplophrentis carinatus*[51,52]. By contrast, the relationship with molluscs has

also been emphasised on the basis of the inner shell structure and biomineralisation of *Allatheca*, *Microcornus*, *Parakorilithes*, *Conotheca* and *Cupitheca*[53,54]. Hyoliths are distinguished from cephalopods by several features (Landing and Kröger[25] and references therein), such as the operculum with its muscles[47,55] or differences in septum morphology[48]. Some orthothecid hyoliths possess chambered apical portions[48], lacking the cephalopod-characteristic connecting ring and siphuncle. The septa dividing the conchs of orthothecids are of variable shapes. They are calcitic, although calcite is commonly replaced by phosphatic material during diagenesis[48]. The internal morphology of these septa

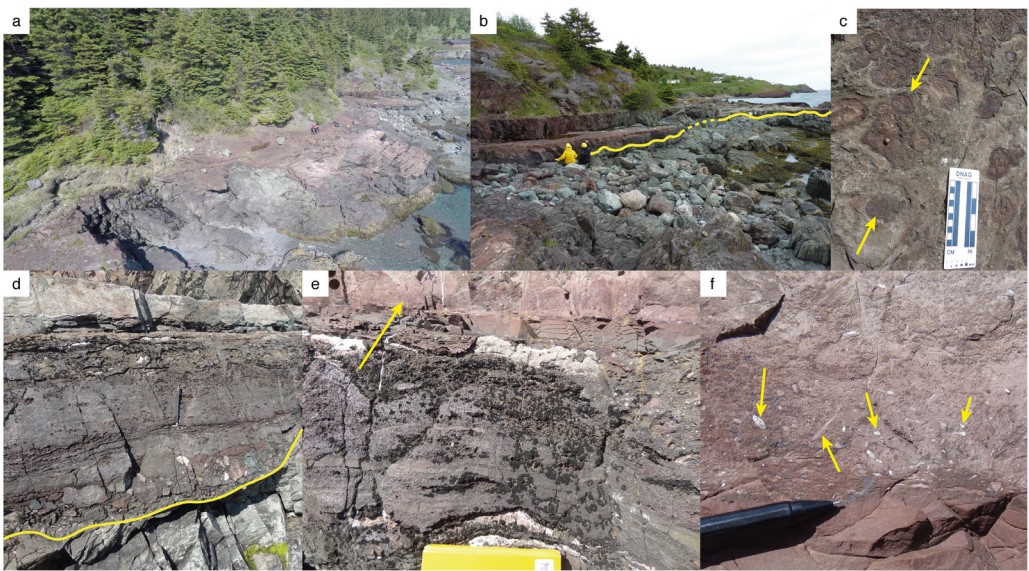

**Fig. 6 Fossil site at Bacon Cove.** Yellow line in **b** and **d** marks the PƐ/Ɛ boundary. **a** Aerial view from NE. Geologists for scale. **b** Outcrop overview. Geologists for scale. **c** Surface of Bonavista Formation hosting abundant oncoids and microbialites (arrows). **d** Contact between uppermost Ediacaran (Drook Formation) and lowermost Cambrian strata (Bonavista Formation) in Bacon Cove. **e** Section, c. 160–200 cm (Bonavista Formation) with arrow marking the coquina layer. **f** Coquina containing the fossil material described here. Arrows mark calcareous shells; view perpendicular to bedding.

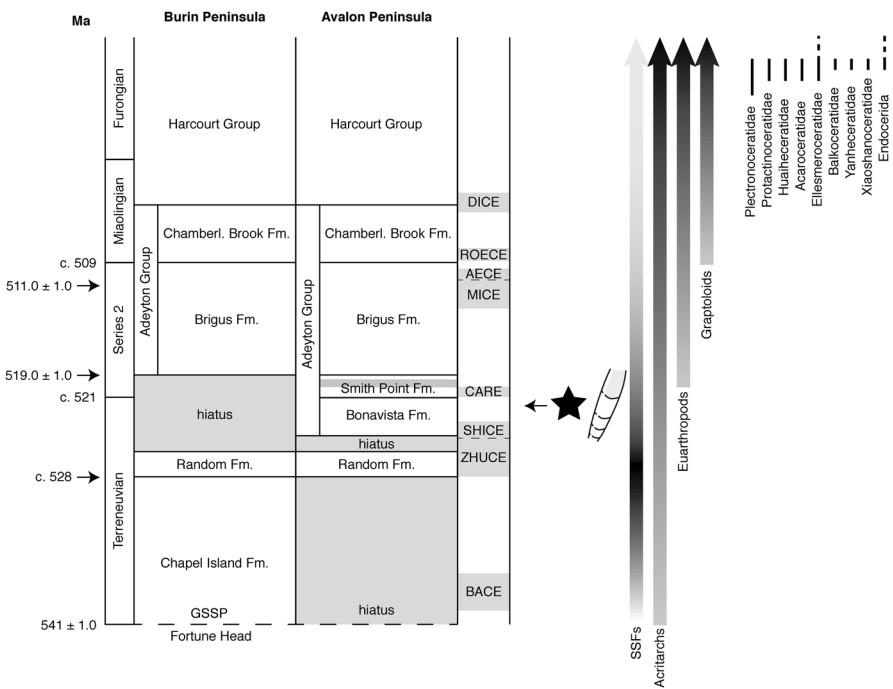

**Fig. 7 Stratigraphic context of the fossil material described here.** Note that the level of occurrence (marked by star) precedes the first occurrence of euarthropods (c. 521 Ma) and Graptoloids, as well as the hitherto known first occurrence of cephalopods (Plectronoceratidae). Abbreviations: Fm Formation, SSFs Small Shelly Fossils. Family range chart from Fang et al.[8]. Stratigraphy from Hutchinson[11], Landing et al.[15], King[18], and McCartney[83]. Geochronology: 530.7 ± 0.9 Ma[87] recalculated as c. 528 Ma[25,88]; 543.9 ± 0.2 Ma, 519.0 ± 1.0 Ma, 511.0 ± 1.0 Ma, 570.94 ± 0.38 Ma, 566.25 ± 0.35 Ma[89]. Composite carbon isotope excursions compiled from Brasier et al.[83] (SHICE, CARE) and Peng et al.[90] (BACE, ZHUCE, MICE, AECE, ROECE, DICE).

consists of two fibrous layers, which are frequently preserved during diagenesis and secondary overgrowth[48]. By contrast, the high-Mg-calcite septa of our material, although diagenetically impacted, do not show any fibrous characters, different layers, nor have they been replaced by phosphatic material (Fig. 1). Thus, the internal structure of our material differs from that of chambered hyoliths, such as *A. degeeri*, and an operculum is also absent.

Whereas *P. cambria* was long regarded as the oldest cephalopod, a variety of undisputed cephalopods are widely accepted to occur in slightly younger stages of the late Cambrian (Stage 10). They have been documented, e.g. from Australia[56], China[10,57–61], North America[34,37] and Siberia[62]. Their morphology varies with respect to shell morphology, septal spacing, septal neck, and connecting ring. Their conchs are either longiconic or

breviconic, curved endogastrically (e.g. Plectronoceratidae, Balkoceratidae, Huaiheceratidae) or straight (e.g. Ellesmeroceratidae, Acaroceratidae). For example, *Dongshanoceras jiangshanense*[61] and all species of the genus *Ectenolites*[63] show longiconic and straight conchs[60,61], similar to our material. An orthoconic shell was recently proposed for the earliest stages of cephalopod evolution[6] and is supported by our material.

The septal spacing of late Cambrian cephalopods is usually small. The ratio of the septal spacing to the corresponding diameter of the shell usually lies between 0.05 and 0.1 in plectronocerids and 0.2 in ellesmerocerids[60]. Our material shows ratios of 0.5–0.6 and thus resembles Ordovician members of Orthocerida, although it should be noted that our measurements may be biased because of oblique sections.

The siphuncles of the earliest known cephalopods are situated marginally[10,57–62]. This is not seen in the specimens documented here, in which the siphuncle is close but not adjacent to the shell wall.

**Diagenetic history**. In our specimens, diagenetic cement accumulated in the hollow parts of the shell. This occurrence is particularly important for the interpretation of specimens nos. NFM F-2776 and F-2777, in which cross sections of the shell show a less marginal position of the siphuncle. In these specimens, calcite spar cement filled the empty chambers during an early diagenetic stage. Only the siphuncle was subsequently filled with muddy sediment. This indicates a connection of the siphuncle with the body chamber, as the infill was probably washed through the open body chamber into the siphuncle connecting the phragmocone compartments. The homogeneous diagenetic history therefore provides strong evidence for the presence of a siphuncle piercing at least some septa.

SEM-EDX maps reveal manganese-calcite enrichments in the phragmocones of specimens nos. NFM F-2774 and NFM F-2776 (Figs. 1d and 2c). Cameral deposits are unusual in early cephalopods, but deposits similar to those identified here have been reported from the Ordovician *Bactroceras*[64]. We interpret the deposits as a species-related feature. Despite the abundance of calcareous shells in our thin sections, none of these show manganese enrichments. This indicates that diagenetic features, such as a bulk-rock-related enrichment with manganese intruded by fluids or similar processes, can be excluded, because then these features would also be present in other fossil shells. Also, there is no evidence of diagenetic alteration affecting only the cephalopod specimens previous to their potential transport. Therefore, we interpret the presence of manganese as a feature specific to the cephalopod specimens documented here, indicating that NFM F-2774 and NFM F-2776 are conspecific.

Three rounded objects are illustrated in Fig. 1a–c on the left side of NFM F-2774 and may represent broken septa, secondarily overgrown by diagenetic deposits. This scenario is supported by Mg-enrichment of these features (Fig. 1c), as all unambiguously identified septa of our material are enriched in Mg. Clearly, this would fit the observation of Chen and Teichert[59] that early cephalopods are characterised by closely spaced septa.

K-Al-Mg-rich clay minerals enriched in the apical shell portions of NFM F-2774 (Fig. 1c, e; Supplementary Fig. 2d) are interpreted as diagenetic replacements of original septal material, as indicated by Mg-enriched traces (Fig. 1c). A Recent nautiloid shell is typically composed of 99.50% $CaCO_3$, 0.16% $MgCO_3$, 0.15% $(Al, Fe)_2O_3$, and 0.19% $SiO_2$, with aragonite as the only carbonate mineral[2]. Calcium phosphate may be present as traces[2]. With respect to its diagenetic history, the shells of our specimens thus resemble the chemical composition of extant nautiloids, although some typical constituents (e.g. magnesium in septa) are only preserved as traces. Measurable amounts of calcium, magnesium, phosphorus and silicon can be expected to occur in the shell material. The diagenetic history of our material thus provides strong evidence for an originally aragonitic shell.

**Faunal assemblage and depositional environment**. The specimens described here were collected from a coquina deposit and may thus have been transported. However, preservation of these thin-shelled fossils is excellent, which suggests a short transport. Alternatively, the concentration of shells in the coquina may reflect condensation resulting from winnowing of fine-grained siliciclastic material during storm events or through low-energy bottom currents. In this case, transport distances may have been even shorter. In addition to the material described here, the coquina also contains disarticulated elements of small shelly fossils (SSFs; Supplementary Fig. 1), but these do not allow for a reliable interpretation of the depositional environment. Nevertheless, the Bonavista Formation is commonly interpreted as a shallow-marine depositional onlap resulting from a NW–SE transgression onto the coast of the Avalonian microcontinent[13,14]. The underlying conglomerate, c. 20 cm below the coquina, and the directly overlying limestone contain in situ preserved, stratiform, columnar stromatolites composed of rivulariacean and epiphytacean algae[12], an association that also suggests shallow-water conditions in the photic zone.

**Possible implications on the origin of the Cephalopoda**. The specimens described here may represent the earliest cephalopods capable of regulating the buoyancy of their shell through a siphuncle. This view supports earlier assumptions that cephalopods originated in the early Cambrian[1,6] and also corroborates the idea of a monoplacophoran ancestry of the group[1,38,40,65–67]. Molecular studies also support a close relationship between Cephalopoda and Monoplacophora[67,68].

The non-siphuncled monoplacophoran *Knightoconus* has been considered as an ancestral cephalopod, although with inefficient buoyancy. It is younger than our material and can therefore be discarded from the list of potential precursors. The present taxon may represent a connecting link between septate non-siphuncled monoplacophorans (e.g. *Tannuella*) and cephalopods, but its similarity with orthothecid hyoliths may also indicate an ancestry with the latter group, as suggested by Dzik[35,69]. However, there are morphological differences between orthothecids and cephalopods[25], and in the present case, the difference in septum morphology and absence of an operculum also opposes an assignment to orthothecids. *Nectocaris pteryx*[70] from the middle Cambrian Burgess Shale, Canada, was interpreted as an early soft-bodied cephalopod by Smith & Caron[71], on the basis of a structure interpreted as a funnel. Based on this interpretation a shell-less coleoid-like ancestor of cephalopods was hypothesised[71,72]. However, *N. pteryx* lacks unequivocal molluscan characteristics, and other features even oppose an assignment to coleoids, such as the presence of lateral fins and camera-like eyes[1]. The axial cavity and the funnel could not have served as jet-propulsion or respiratory systems in *N. pteryx*, due to their shapes. *Nectocaris* may therefore represent an independent group of Lophotrochozoa[1]. The material described here clearly resembles a shelly cephalopod and thus contradicts an early Cambrian *Nectocaris* ancestry of cephalopods as well.

Another important difference of the present material and later cephalopods, or the Ordovician *Bactroceras*[31], refers to the connecting ring which in the present material appears to have consisted of relatively soft phosphoric material. It is possible that the soft siphuncle marks a transitional step between non-siphuncled monoplacophorans and cephalopods possessing a

stable carbonate siphuncle. Mutvei[6] recently suggested that the earliest cephalopods may have presented a non-mineralised connecting ring and that the siphuncle may have evolved from a single septum providing osmosis. The non-mineralised phosphoric connecting ring interpreted here to be present in our material may point towards this direction, although there is as yet no strong evidence for this interpretation.

Our material may extend the origin of cephalopods to the Terreneuvian, before the Cambrian Arthropod Radiation Excursion (CARE[73]). It will then predate the undoubted cephalopod *Plectronoceras cambria* by about 30 million years. Researchers have pointed to the gap between the origin of other high-level taxonomic groups and the earliest occurrence of cephalopods[25], a gap which may now be closed by our specimens. In this case, the origin of cephalopods will precede the global onset of major groups such as euarthropods and graptolites (both c. 521 Ma[74,75]) and possibly coincide with the peak in diversity of SSFs.

It is intriguing to note that cephalopods, the highest organised molluscs, were present in this early stage of metazoan radiation but remained almost unperceived in the fossil record for ~30 million years. While authors have referred to the development of the siphuncle as an accident, leading to unforeseen evolutionary success[66], the development of a phosphoric soft connecting ring as identified here, may have represented an evolutionary dead end, as the fossil record does not present further evidence for this morphology, except for *Bactroceras latisiphonatum*[76]. It has been suggested that this Ordovician ellesmerocerid, with its phosphatic siphuncle providing stability, was capable of occupying deep-water shelf habitats[31]. Likewise, the soft siphuncle identified in our material might have led to a restriction to extremely shallow-water habitats, a hypothesis supported by the depositional environment of the host rock at Bacon Cove. Potential descending taxa either inhabited more shallow environments or became extinct as the Bacon Cove habitat gradually deepened. Unfortunately, shallow-water deposits were not preserved on the Avalon Peninsula until the early Ordovician, from which unequivocal cephalopods have been described[77].

All Cambrian cephalopods known to date reveal a restriction to low latitudes[8], while Avalonia was located in a temperate to southern high-latitude position[78]. It is thus remarkable that early Cambrian cephalopods may have evolved under environmental conditions different from those of late Cambrian representatives. In addition, they may have been restricted to Avalonia. Endemism is indicated by the absence of material similar to ours in adjacent terranes, such as Ganderia, but also in Cambrian Lagerstätten with soft-body preservation. It appears to us that characteristic features such as a phosphoric siphuncle should have been preserved in these exquisite deposits.

**Unequivocal cephalopod identity?** We are aware that an unequivocal cephalopod assignment of the specimens described and discussed here must await future findings of better-preserved material less affected by diagenesis. For example, no siphuncle was established in NFM F-2774 and uncertainties thus remain about the conspecificity of this specimen and others presented here (e.g. NFM F-2776). Where a siphuncle is identified, it is situated peripherically, but not as marginally as expected, and it may originally have consisted of soft phosphorous material. Our interpretation of cameral deposits is at odds with their absence in other Cambrian cephalopods. The biologic origin of these deposits is also uncertain as manganese is contained within the diagenetic cement; this condition differs from cameral deposits in other Palaeozoic cephalopods[79–82]. Thus, the manganese-bearing deposits may not be cameral deposits *sensu stricto* but biological

enrichments of uncertain origin or, although unlikely, are diagenetic features.

The phosphorous-bearing outer and calcitic inner walls of the potential connecting ring in NFM F-2776 (Figs. 2 and 3) are also remarkably unusual features in fossil cephalopods, as siphuncle and connecting ring structures should be homogeneous by themselves[29]. We can therefore not exclude a scenario in which phosphorus enrichment resulted from cone-in-cone processes, in particular as SSFs presenting these structures are present in the limestone layer investigated here and are readily identified in cross sections of individuals of this assemblage. Nevertheless, the potential cephalopod described here, including NFM F-2776, is easily distinguished from these SSF elements by a spar envelope around the apparent siphuncle. Also, a scenario invoking cone-in-cone processes would be complex and include the following stages: (1) deposition of the original specimen, (2) infill of another conical fossil already filled with sediment, whereas the original specimen is still devoid of sediment, and finally (3) sealing of the entire composition during diagenesis. This scenario would still not explain the phosphoric ring identified in the interior of NFM F-2776 (Fig. 3). Therefore, this latter scenario requires an additional prerequisite, i.e. soft sediment rich in P, Ca and possibly Mg accumulating in the specimen and there precipitating the above elements in circular shape around the washed-in specimen, which was later destroyed by diagenesis. Typical matrix elements such as K, Al and Si are absent in this part of NFM F-2776. Clearly, this alternative scenario requires numerous assumptions. Applying Occam's razor, we therefore favour the interpretation that the phosphorus enrichment is evidence for a connecting ring.

The present material clearly differs from typical early Cambrian septate molluscs, hyoliths and other SSFs, e.g. in septal morphology and shell structure, and was therefore not assigned to a known early Cambrian taxon. However, a failed assignment to a known taxon combined with uncertain cephalopod features, does not constitute unequivocal cephalopod evidence. We therefore provisionally allocate a cephalopod identity of our specimens, pending better-preserved material, which would have extensive phylogenetic, stratigraphic, palaeogeographic and morphological implications. Based on the present data we suggest that future search for early and middle Cambrian cephalopods should focus on orthoconic fossils with close septal distances, a connecting ring and a siphuncle preserved as soft, phosphorous material.

**Conclusions**

The material from the early Cambrian of the Avalon Peninsula, Newfoundland, arguably represents the earliest cephalopod known to date. It is characterised by a straight, elongate, conical shell, with a potential siphuncle in a peripheral, although not explicitly marginal position. The taxon inhabited an extremely shallow, marine environment. The presumed cephalopod may have originated from non-siphuncled monoplacophorans, but the development of a primitive non-mineralised siphuncle from a single septum is also possible. Coleoid-like shell-less forms or orthothecid hyoliths as cephalopod ancestors appear unlikely, if a cephalopod identity of the material will be confirmed by future studies.

**Methods**

Scanning electron microscopy (SEM) was conducted with a ZEISS EVO MA 15 SEM. The Energy Dispersive Spectroscopy (EDS) detector was a silicon drift detector (SDD) X-MaxN 150 mm$^2$ by Oxford Instruments. EDS mappings were analysed with AZTEC 4.2. Thin sections were prepared from c. 30 µm thick slices and the microscopical analysis conducted with a Keyence VHX-6000 digital microscope. All specimens are housed in the Provincial Museum Division, The Rooms Corporation of Newfoundland and Labrador, St. John's, Newfoundland, Canada (NFM).

**Reporting summary**. Further information on research design is available in the Nature Research Reporting Summary linked to this article.

## Data availability
All specimens are housed in the Provincial Museum Division, The Rooms Corporation of Newfoundland and Labrador, St. John's, Newfoundland, Canada (NFM). The authors declare that the data supporting the findings of this study are available within the article and its supplementary information files.

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

## Acknowledgements

The authors thank the Klaus Tschira Foundation (Grant 00.272.2015) and the ODWIN gGmbH for generous financial support. The Province of Newfoundland and Labrador and the Manuels River Natural Heritage Society are thanked for logistical support. We thank Stuart Crosbie (Newfoundland), Linda Fischer, Melanie Kling and Tanja Unger for their assistance during the fieldwork, Christina Ifrim for conducting grinding tomography, Alexander Varychev for assistance with SEM analyses and Pascale D. Emondt (all Heidelberg University) for assistance with the illustrations.

## Author contributions

The study was designed by A.H., G.A. and W.S. and is based on field work carried out by A.H. and G.A. Palaeontological interpretations and implications developed from discussions with D.F., W.S. and P.B. A.H. and G.A. were the main authors and contributed equally to the work. The manuscript was prepared with input from all co-authors.

## Funding

## Competing interests

The authors declare no competing interests.
