## [Peer Review File · Communications Biology]

Reviewers' comments:

Reviewer #1 (Remarks to the Author):

Dear Dr. Hildenbrand and co-authors,
Dear Dr. Grinham,

I reviewed the article "Unexpected and highly evolved: A cephalopod from the early Cambrian" by Anne Hildenbrand and co-authors.

In this article, the authors claim to present the interesting first evidence for early Cambrian cephalopods, pushing back the origin of cephalopods by more than 20 million years. If correct, this will certainly be highly relevant not only to cephalopod researchers, but also for our understanding of the evolutionary history of the Mollusca, the Cambrian Explosion and the diversification of early metazoan life. That being said, I am not convinced by the cephalopod identity of the species (details are given below). Furthermore, even if the cephalopod identity would be accepted, the preservation and preparation of the specimens essentially prevents any comparison with established Cambrian cephalopod species. This means that the only way to refer any specimens to the new species and genus seems to be its age, because its diagnostic characters are held so general that they could just as well apply to a large number of Cambrian and Ordovician species. Lastly, the authors did not include important references on Cambrian cephalopods.

My suggestion is that the authors should consider more the possibility of alternative explanations or otherwise substantiate their claim that these specimens are really cephalopods. The comments below may give you an idea what would be needed for this. Furthermore, I would advise against the erection of new taxonomic names based on the material presented. Unfortunately, I do not know whether the material would be unusual enough to warrant publication in *Communications in Biology* if they are not considered cephalopods, but perhaps a discussion of the pros and cons of different alternative taxonomic assignments (including cephalopods) of the specimens would be useful. However, I am aware that this requires a major restructuring of the manuscript that may not be desirable. Nevertheless, the evidence that *Avaloniaceras* is a cephalopod is far from unequivocal given the current state of the manuscript and I would rather avoid another "Nectocaris-situation", where another Cambrian animal with unclear affinities was claimed to be the oldest cephalopod which was subsequently refuted by most cephalopod workers.

Detailed comments:

As already stated above, I am not convinced that the specimens are cephalopods. Particularly in cephalopods, there have been repeated claims of older representatives, all of which turned out to be wrong, e.g. *Volborthella*, *Salterella*, *Vologdinella*, *Nectocaris* – the former three examples show that organisms with perforate "septa" are present in the early and middle Cambrian, but they are not necessarily cephalopods. In my opinion it is unlikely that *Avaloniaceras* is a cephalopod for the following reasons:

1) You present two specimens in longitudinal sections that exhibit septa, and two other specimens in cross section that have a circular structure near their shell margin, interpreted as siphuncle. However, it is not clear to me, how you come to the conclusion that these specimens belong to the same species – after all, the former two show no sign of a siphuncle. The siphuncular wall is mentioned in the captions of Fig. 1, but this is missing in the description and I can see nothing in the holotype that looks like a siphuncle. At the same time, the two other paratypes show no traces of septa. In cross section, the septa should be visible as concentric rings, but considering the wide septal spacing, it may be possible that the shell was sectioned at a position where no septa are present. Another thing to consider is the somewhat irregularly distorted outline of the cross section, while the longitudinal section seems to be an almost perfect cone (but note that the septa seem to be deformed, they are not evenly concave as would be expected in a cephalopod). Obviously, it is possible that preservation plays a role here, the authors will probably know best, whether deformation of fossils is common in the type horizon. Maybe I am overlooking something, but you should support better your claim that the specimens are conspecific.

2) Assuming that the specimens belong to the same species there are still some other issues. As stated above, there are other early-mid Cambrian organisms that have "perforate septa". The siphuncle of all cephalopods consists of the septal necks and the connecting ring. Unless either of these structures are clearly shown, a cephalopod affinity or identity must remain doubtful, because it is the siphuncle - a very specific and unique structure within molluscs - that makes a cephalopod, and not aragonitic septa. Unfortunately, the material of Avaloniaceras shows no such evidence. On the contrary, as far as I can judge, there is no evidence of a connecting ring (siphuncular wall in the authors terms) in the cross section of the first paratype in Fig. 1, but this should be visible with this type of preservation. Otherwise, if the connecting ring was destroyed during its early depositional history (either mechanical or by implosion), matrix would have been able to enter the chambers, or the calcite spar would have filled the entire space of the chamber, leaving no trace of the siphuncle in cross section except at the septum (i.e. septal neck). As discussed above, this is not the case here, because the septa are not visible. In the other cross-sectioned paratype in Fig. 2, there is a "siphuncular wall" visible, but this looks very similar to the outer shell wall and I do not think that this is the connecting ring. This issue could potentially be resolved by additional material where the possible siphuncle is exposed in longitudinal sections.

3) The species appears to be very different from other (late) Cambrian cephalopods. While this is not an issue per se, it raises at least suspicions and opens up questions as to why there was apparently such a drastic change in morphology. The most important differences are:

- Avaloniaceras has a relatively wide septal spacing. Late Cambrian cephalopods have extremely closely spaced chambers, usually about 0.05-0.1 in plectronocerids and below 0.2 in ellesmerocerids when compared to the corresponding shell diameter (Chen & Teichert 1983). You give a value of 0.6 for Avaloniaceras, but from the measurements and figures it becomes evident that this varies between 0.5 and 1.0, increasing during ontogeny (see also comment further below). The first cephalopods with similar (adult) chamber lengths are members of the Orthocerida, but they do not occur until the Ordovician. In my experience, relative chamber lengths above 0.3 are very rare before the Middle Ordovician. It has been argued that the close septal spacing helps counterbalancing buoyancy (e.g. Crick 1988), thus, Avaloniaceras would require alternative mechanisms that prevent it from floating to the surface. The other thing is that relative chamber length usually decreases during ontogeny. To my knowledge, there is no general study on this, but a similar trend has been shown in numerous taxa (e.g. Evans 2005; Kröger 2011; Aubrechtová 2015; personal observations in Cambrian taxa). The only exception is at the earliest ontogenetic stages, where there appears to be more variation (compare e.g. Kröger & Keupp 2004; Evans 2005). Lastly, the authors explicitly cite the exponential growth of septal spacing as an argument for a cephalopod identity; however, in the cited article (Tajika et al. 2020) it is shown that chamber volume increases exponentially. Because of allometry, absolute septal spacing (length!) will increase more linearly, if volume increases exponentially. Therefore, septal spacing does not corroborate a cephalopod identity, but rather speaks against it. It may be argued that the specimens represent juveniles, but they would still be unusual in comparison with Cambrian cephalopods and I am unsure if much larger organisms could be expected in this assemblage. Another problem is that Tajika et al. (2020) refer exclusively to coiled shells, while Avaloniaceras is straight.
- Virtually all Cambrian cephalopods have a marginal siphuncle, which means that the siphuncle is touching the shell wall. Exceptions are very likely due to the section plane (note that this was not recognized by Chen & Teichert 1983; this argument is based partly on Wade & Stait 1993 and further unpublished personal observations). I have never seen any specimen or cross section of a Cambrian cephalopod where the siphuncle is demonstrably as far removed from the shell wall as in your specimen (i.e. more than its own diameter).
- The shape and position of the siphuncle cross section appears to vary extremely between the two cross-sectioned paratypes, based on Fig. 2 and your measurements in supplementary Fig. 6. The paratype in Fig. 2 has an extremely laterally compressed siphuncle cross section that is somewhat displaced compared to the symmetry axis. Usually, the siphuncle has a more or less circular cross section, but in that paratype the width/height ratio is apparently about 0.4, which is highly unusual and the only thing even remotely similar occurs in very specialised forms such as the piloceratids (the lowest I have seen is around 0.65).
- The species appears to be perfectly straight. Cambrian cephalopods are almost always at least slightly curved endogastrically or one side is more convex than the other, usually the dorsum. This may not be a significant difference, but it simply adds up.

4) You do not discuss alternative explanations. I have no expertise on SSFs, so I am in no good position to suggest any specific taxa, but perhaps this "siphuncle" could result from a small specimen washed into the body chamber of a larger specimen, much in the same way as in the holotype? The "siphuncle" in Fig. 2 looks quite similar to the surrounding small fossils. Note that there is a nice little paper on a similar phenomenon called "telescoping", which may be useful to cite (Hladil et al. 2014).

Another issue with the manuscript is that it ignores a large part of the existing literature dealing with Cambrian cephalopods and the only comparisons are made with *Plectronoceras*. While this genus is currently the oldest and you correctly state that it is poorly known (but arguably from much better material than *Avaloniaceras*!), only slightly younger cephalopods are known from rich assemblages in the latest Cambrian (Stage 10) of China, North America, Australia, Siberia and possibly Kazakhstan (Flower 1954, 1964; Chen et al. 1979a,b; Chen & Qi 1982; Chen & Teichert 1983; Li 1984; Wade 1988; Wade & Stait 1993; Mutvei et al. 2007; Fang et al. 2019; Dzik 2020). Chen & Teichert (1983) is probably the most important reference for Cambrian cephalopods and at least this should be cited somewhere. Amongst other things, it shows that starting with *Plectronoceras*, there is a continuous record of cephalopods until the latest Cambrian. While there has probably been a tendency to split and some of the species are probably based on oblique sections, it provides a good overview on the morphological range of Cambrian forms. At least some of these should be compared to the described material. Currently, most of the references cited by the authors are not dealing primarily with Cambrian cephalopods.

While you claim that the specimens are "well preserved" I would argue that the material hardly justifies the establishment of a new genus and species, at least not in its current state of preparation. The descriptions are based on sections that are likely oblique and off the symmetry plane, making it impossible to recognize crucial details of the siphuncle that would serve to distinguish it from any other early Palaeozoic cephalopod or even clearly identify it as one. The generic description in its current state would probably fit hundreds of species thus far described from the Cambro-Ordovician, while at the same time not mentioning the differences outlined above. The measurements are also doubtful and likely biased because the section is not in the symmetry plane. For example, the first chamber is measured from the apical end to the first internal septum. However, the apical end probably does not represent the last preserved septum, but rather where the section met the outer shell wall. Only if the symmetry plane is at least approximately known or at least the structure of the siphuncle would I consider that these specimens are worth the establishment of a new taxon, at least if cephalopod standards were to be applied. I therefore strongly advise against the description of a new species, at most *Ellesmerocerida* (?) indet. would be acceptable, if you insist on a cephalopod identity (it lacks any characters that would allow assigning it to the *Plectronocerida*). At the moment, I would also avoid the use of subclasses, because they may change in the future depending on the development of the new Treatise.

Some further minor comments:

Title: I am not sure what you mean by "highly evolved"; the species presented here is in no way more "advanced" than other Cambrian or Early Ordovician cephalopods.

Line 11: "c. 531 Ma": This seems a bit awkward, nowhere has a precise date for the appearance of cephalopods been proposed and if it was, it is pure speculation without any evidence. If you insist on leaving a date, I suggest you either round to "c. 530 Ma" or state a range within the early Cambrian. However, I recommend leaving out a date entirely, because this is essentially just a wild guess that cannot be falsified.

Line 23-24: "Squids, cuttlefish, and octopuses originally possessed a chambered shell as the pearly *Nautilus* impressively demonstrates": This sentence is not entirely correct: 1) *Sepia* and *Spirula* are examples of recent coleoids that still possess a chambered shells, but in contrast to *Nautilus*, they are internal. 2) This may seem a bit pedantic, but it were not the squids, cuttlefish and octopuses themselves that originally possessed an (external) chambered shell, but their ancestors!

Line 25: "probably benthic precursors": Citing Yochelson, Flower & Webers (1973) or Webers & Yochelson (1989) would be more appropriate here, even if you don't agree with their Knightoconus hypothesis. Alternatively, you may cite Kröger, Vinther & Fuchs (2011).

Line 31-33: "In members of the related order Ellesmerocerida, the shell is divided into chambers by concave-shaped septa, and the siphuncle is relatively wide and located marginally along the ventral side of the phragmocone": This is also the case in the Plectronocerida – in fact, apart from the wide ventral siphuncle, all characters that you mention are also found in a majority of Palaeozoic cephalopods.

Line 51: "Shell breviconic": the term "breviconic" is usually applied to species with high expansion rate, I would therefore consider this specimen as "longiconic" (alternatively, you could state the expansion rate/angle directly, because this is less subjective). Furthermore, both brevi- and longiconic species can be slightly curved (e.g. Plectronoceras could also be considered as breviconic). Therefore, it is important to state the curvature, i.e. in this case either "straight" or "orthoconic".

Line 81: "cone angle of 14-24°": Keep in mind that the angle of expansion (this would be the correct term) is highly dependent on the section plane. If the section is oblique, the angle would be highly biased. The rounded anterior and posterior ends imply that the section is indeed oblique or off the symmetry plane (a way to experiment with this is by drawing a cone in a 3D drawing or geometry program and manually producing sections in different orientations and comparing their shapes).

Line 91: "implicating a saggital rather than a longitudinal section": As above, the section is not necessarily parallel to the median plane, but can also be oblique, thus biasing measurements. This is already a very serious problem for the large number of Chinese Cambrian taxa, which are almost exclusively known from thin sections. If further specimens are found, it will be difficult to tell whether any variation is intraspecific, due to species-differences or simply due to differing orientations of the section planes.

Line 92: "It is possible that the apertural and apical ends of the shell are missing in the specimen.": Or probably outside the section plane.

Line 99: "4.6 mm long and 3.6 mm wide": When using directions in a cephalopod, any measurement in dorsoventral direction is "high", in lateral direction "width" and in longitudinal direction "length". Accordingly, it would be "high" instead of "long" here. Please check the manuscript carefully for similar mistakes, this can be very confusing to read.

Line 152-174: Diagenesis: I am not familiar with SEM-EDX, so I cannot judge this part. However, it might be useful to cite similar studies on cephalopod or other originally aragonitic shells, if possible.

Line 176-184: Septal distance: I already explained my critique to this point above. I do not understand why you first state that chamber volumes can easily be estimated but then use only chamber length anyway. I wonder what would come out if you did the same analysis on another septate Cambrian mollusc. Would you expect this to differ from exponential growth? Also, I do not think that only four data points (out of which the first is questionable because it probably represents the shell margin rather than a septum) are statistically sufficient to support your conclusion.

Line 202-214: This paragraph is held quite short, although the implications are quite far-reaching. It would mean that cephalopods evolved much earlier than expected (Plectronoceras is known from the late Jiangshanian, so the difference is probably closer to 30 million years instead of 20 as you suggest). Cephalopods have always been thought to undergo an explosive radiation in the latest Cambrian because they were able to colonize the free water column, thus occupying new niches. Your find would refute this hypothesis, because they would have been around for 30 million years in which they were apparently extremely rare. This leads to the question why it took them so long and what triggered the radiation in the latest Cambrian. The discussion on cephalopod

ancestors is held quite short and no arguments are presented in relation to the evolution of the siphuncle, which is probably the most important open question in the search of the cephalopod ancestor. The research history of this subject is quite complex, see Webers & Yochelson (1989) for a detailed discussion of various hypotheses. Mutvei (2020) recently suggested yet another hypothesis which is highly speculative, but he assumes that simple siphuncular structures are inherently ancestral compared to complex ones, which I think is not necessarily the case. Another aspect that needs to be discussed in this paragraph is that according to Kröger (2013) and Fang et al. (2019), late Cambrian cephalopods are exclusively restricted to tropical palaeolatitudes. I am not familiar with the palaeogeography of Newfoundland, but if it fits also into this tropical scenario it would definitely be worth mentioning. Otherwise, this would suggest a high latitude origin of cephalopods with subsequent migration and restriction to tropical regions. I do not know how likely this scenario would be.

Figs.: The orientation of the cross sections is upside-down in all figures. The side with the siphuncle is generally regarded as the ventral side, therefore, this is supposed to be at the bottom.

Supplementary Fig. 2: The family range charts on the right side are directly taken from Fang et al. (2019) and should be cited!

If you have any questions, please feel free to contact me and I will be glad to help. Also, let me know in case you need specific literature, I am aware that some of these publications may be a bit hard to get.

Best wishes,

Alexander Pohle
University of Zurich

References:

- Aubrechtová 2015, *Geobios* 48: 193–211.
Chen & Qi 1979, *Acta Palaeontologica Sinica* 21: 392–403.
Chen & Teichert 1983, *Palaeontographica Abteilung A* 181: 1–102.
Chen, Tsou, Chen & Qi 1979a, *Acta Palaeontologica Sinica* 18: 1–24.
Chen, Zou, Chen & Qi 1979b, *Acta Palaeontologica Sinica* 18: 103–124.
Crick 1988, *Senckenbergiana lethaea* 69: 13–42.
Dzik 2020, *Acta Palaeontologica Polonica* 65: 149–165.
Evans 2005, *Monograph of the Palaeontographical Society* 158, 1–81.
Fang, Kröger, Zhang, Zhang & Chen 2019, *Palaeoworld* 28, 51–57.
Flower 1954, *New Mexico Bureau of Mining & Mineral Resources, Bulletin* 40: 1–51.
Flower 1964, *New Mexico Bureau of Mining & Mineral Resources, Memoir* 12: 1–164.
Hladil, Simcik, Ruzicka, Kulaviak & Lisy 2014, *Lethaia* 47: 376–396.
Kröger 2012, *GFF* 134: 115–132.
Kröger 2013, *Geological Society of London, Memoirs* 38: 429–448.
Kröger & Keupp 2004, *Lethaia* 37: 439–444.
Kröger, Vinther & Fuchs 2011, *BioEssays* 33: 602–613.
Li 1984, in: *Nanjing Institute of Geology and Palaeontology (ed.), Stratigraphy and Palaeontology of Systemic Boundaries in China, Cambrian-Ordovician Boundary (1): 187–265.*
Mutvei 2020, *GFF* 142: 115–124.
Mutvei, Zhang & Dunca 2007, *Palaeontology* 50: 1327–1333
Tajika, Landman, Hoffmann, Lemanis, Morimoto, Ifrim & Klug 2020, *Scientific Reports* 10: 2950.
Wade 1988, *New Mexico Bureau of Mining & Mineral Resources, Memoir* 44: 15–25.
Wade & Stait 1993, in: *Beesley et al. (eds.), Mollusca: The Southern Synthesis, Part A: 485–493.*
Webers & Yochelson 1989, *Geological Society Special Publication* 47: 29–42.
Yochelson, Flower & Webers 1973, *Lethaia* 6: 275–309.

Reviewer #2 (Remarks to the Author):

Dear authors,

congratulations on the description of such an interesting material!

The submitted study describes early Cambrian fossils from Newfoundland interpreted as cephalopod shells. The holotype is obliquely cut and the presence/absence of the siphuncle thus cannot be unequivocally proven. However, since the paratypes look very much like cephalopod shell cross-sections with the siphuncle, I am tempted to agree that the fossils indeed represent the oldest cephalopods. As such, the study results are novel and, if published, will be of interest not only for specialists working with both fossil and extant cephalopods but also for researchers studying various aspects of early Paleozoic diversifications of life.

The text is well-structured and generally clearly written. I can see no major linguistic or formal issues. The used methodology in addition to classical palaeontological methods supported the conclusions of the work. In my opinion, the main flaws of the paper are related to insufficient citations and morphological descriptions of the specimens. Following are more specific comments and suggestions.

1. On line 23, I would suggest a small addition to increase accuracy of the statement: The ANCESTORS of squids, cuttlefish, and octopuses...

2. Line 26 – please add the reference number 1 (Kröger et al. 2011) after “precursors”.

3. Lines 28 – 33 – I strongly recommend to add some more citations. It is especially not sufficient to cite only the Treatise and a single other work here. The publications to consider particularly are: KRÖGER, B., SERVAIS, T. & ZHANG, Y. 2009. The Origin and Initial Rise of Pelagic Cephalopods in the Ordovician. *PLoS ONE* 4, e7262.

KRÖGER, B. & ZHANG, Y. 2009. Pulsed cephalopod diversification during the Ordovician.

Palaeogeography, Palaeoclimatology, Palaeoecology 273, 174–183.

KRÖGER, B. 2013. Cambrian–Ordovician cephalopod palaeogeography and diversity, 429–448. In HARPER, D.A.T. & SERVAIS, T. (eds) *Early Palaeozoic Biogeography and Palaeogeography*. Geological Society, London, *Memoirs* 38.

FANG, X., KRÖGER, B., ZHANG, Y.D., ZHANG, Y.B. and CHEN, T.E., 2019. Palaeogeographic distribution and diversity of cephalopods during the Cambrian–Ordovician transition. *Palaeoworld*, 28(1-2), pp.51-57.

4. In the Introduction, a similarity between Avaloniaceras and the Ellesmeroceratida is implied. However, the new genus is assigned to the Plectronoceratida with uncertain classification on the ordinal level. This may be confusing to the reader. Please, make clearer to what group you assign your new genus and why.

5. I have some concerns regarding the morphological descriptions of the specimens arising from the fact that the holotype is obliquely cut, out of the plane of symmetry. Under such circumstances, it is not possible to reliably measure the apical angle and it is also difficult to determine the shell shape. Besides that, when the shell is compressed in cross-section, there is not a single value of the apical angle but two because the shell expands more rapidly dorso-ventrally and less rapidly laterally. Please, kindly consider rephrasing.

6. Also, I recommend a slightly more careful interpretation regarding increasing the chamber length along the shell. I am especially skeptical about whether the holotype really has such a long last phragmocone chamber. I would recommend double-checking whether there cannot be some septa secondarily missing. The photograph at Fig. 1a shows three rounded objects on the left side of the shell – can they be secondary deposit overgrowing remains of broken septa?

7. Why do you think the siphuncle was located ventrally?

8. Lines 99 + 105 – I suggest to express the dorso-ventral and lateral shell diameters also as relative values (i.e. compression rate).

9. Please, include some more stratigraphic works, such as:

FLETCHER, T. P., 1972, *Geology and Lower to Middle Cambrian trilobite faunas of southeast*

Avalon, Newfoundland, Ph.D. thesis, University of Cambridge, Cambridge.
BENUS, A. P., and LANDING, E. 1984, Depositional environments and biofacies of the Bonavista Formation (Early Cambrian, eastern Newfoundland), Geol. Soc. Am. Abstr. Progr. 16:3.
LANDING E. (1992) Lower Cambrian of Southeastern Newfoundland. In: Lipps J.H., Signor P.W. (eds) Origin and Early Evolution of the Metazoa. Topics in Geobiology, vol 10. Springer, Boston, MA
O'BRIEN, S.J. and KING, A.F., 2002. Neoproterozoic stratigraphy of the Bonavista Peninsula: preliminary results, regional correlations and implications for sediment-hosted stratiform copper exploration in the Newfoundland Avalon Zone. Current Research. Newfoundland Department of Mines and Energy, Geological Survey, Report, pp.02-1.

Overall, I was happy to read the interesting work and hope to see it published.

Best wishes,
Martina Aubrechtová

Reviewer #3 (Remarks to the Author):

Review of: Unexpected and highly evolved: A cephalopod from the early Cambrian by Hildenbrand et al.

Comments for editors and authors

The paper reports the new finding of fossil shells with internal septa and siphuncles from early Cambrian rocks in Newfoundland, Canada. The fossils are new and interpreted by the authors as the oldest known cephalopod shells. This discovery and interpretation is highly important although also slightly provocative.

Reports of cephalopods predating the established earliest plectronoceratids from the middle Furongian have been published before but have generally been unconvincing. Based on the illustrated material in the new manuscript and the interpretation, it is clear that this report will also be regarded with high interest and be closely examined and probably criticized. That said, the authors do present an interesting case that should be published after some revisions.

One of my main concerns are that the background literature referred to is sometimes slightly outdated. There exist a number of papers on Cambrian cephalopods that should be referenced in the introduction and discussion. I have added some specific notes in the enclosed pdf file.

Another, and more important issue is the fossil material at hand. The illustrated specimens are all from thin sections of bulk rock samples and only a single section of each specimen is available. I realize that the nature of the material may make this difficult to accomplish, but more information would make the story much more viable. Have you tried to get 3D images through CT scanning of the blocks? It may not work if the density contrast is too low, but it may be worth trying.

Alternatively, 3D models could be constructed through serial sectioning of some of the specimens. Compare the excellent results derived from the Silurian Herefordshire biota where the individual fossils are of similar size. I know this is a destructive technology, but it should work, and if you could show in a 3D model that the siphuncles cross multiple septa, no one would be able to doubt your interpretations. As it is now, you rely heavily on inferences from your interpretation of the diagenetic history, which will not convince everyone.

I am also interested to learn more about your interpretations of the structures of the septa and siphuncles, particularly in comparison to what is known on this from the Cambrian Ellesmocerids and Plectronocerids. This would be interesting for any evolutionary discussion.

In the comparison with other Cambrian shells, I see little reason to include Coleoides. It is very clearly very different. However, you should instead expand the comparison to Allathea degeeri to include orthothecid hyoliths more broadly. Many such taxa are septate and otherwise similar to cephalopod shells. In addition, some taxa are known to have preserved soft parts, so you have a base for more detailed discussion here, not least since this has been a hot research topic in the last 3-4 years. Also note that despite the fact that these fossils have recently been referred to the Lophophorates, other authors have strongly countered this interpretation and proposed a molluscan affinity for hyoliths.

You mention the Nectocaris controversy, but are quick to discount it by a simple reference. I think the issue would benefit from a longer treatment in a paper like yours and I would recommend a

summary of the arguments and your own evaluation of the evidence.

And finally you end the paper by pointing out the 20 million year gap between your fossils and the first well established cephalopods. Maybe you can discuss the possible reasons or implications for this situation? It will undoubtedly be discussed in the future. Not least since the late Cambrian cephalopod faunas are so extremely diverse. Perhaps you can also compare to the fossil record of the other molluscan Classes and perhaps the hyoliths again?. At least the bivalves have a somewhat similar Cambrian history.

Otherwise, the paper is very interesting and well written. The illustration are good, although I would like to see 3D images as outlined above. I look forward to see the revised version published in the future.

Author's response to Reviewers' comments

Our manuscript COMMSBIO-20-1975-T entitled “Unexpected and highly evolved: A cephalopod from the early Cambrian” has been reviewed by Alexander Pohle (Reviewer 1), Martina Aubrechtová (Reviewer 2) and an anonymous Reviewer (Reviewer 3), to whom we like to express our gratitude. They perceived several weaknesses in the current form of our manuscript, to which we respond below. The Referees comments are highlighted in blue, each answer is beneath.

Reviewer 1 (Alexander Pohle)

You present two specimens in longitudinal sections that exhibit septa, and two other specimens in cross section that have a circular structure near their shell margin, interpreted as siphuncle. However, it is not clear to me, how you come to the conclusion that these specimens belong to the same species – after all, the former two show no sign of a siphuncle. The siphuncular wall is mentioned in the captions of Fig. 1, but this is missing in the description and I can see nothing in the holotype that looks like a siphuncle. At the same time, the two other paratypes show no traces of septa. In cross section, the septa should be visible as concentric rings, but considering the wide septal spacing, it may be possible that the shell was sectioned at a position where no septa are present. Another thing to consider is the somewhat irregularly distorted outline of the cross section, while the longitudinal section seems to be an almost perfect cone (but note that the septa seem to be deformed, they are not evenly concave as would be expected in a cephalopod). Obviously, it is possible that preservation plays a role here, the authors will probably know best, whether deformation of fossils is common in the type horizon. Maybe I am overlooking something, but you should support better your claim that the specimens are conspecific. Assuming that the specimens belong to the same species there are still some other issues. As stated above, there are other early-mid Cambrian organisms that have “perforate septa”.

We agree, that concentric rings should be visible in cross sections of cephalopod material. However, depending on the position of the cross section, the septa may not be visible, as Reviewer 1 also remarked.

We experienced slight deformation of shells in all of our material. Thus, we account the deformation of septa and outer shell of the specimens to the general taphonomy. Please see lines 151–158 and 168–179 for a detailed discussion on the septa of our specimens.

In addition, we've added the following paragraph to the discussion (lines 214–224):

“SEM-EDX maps reveal manganese-calcite enrichments in the phragmocones of specimens no. NFM F-2774 and F-2776 (Figs 1d, 2c). Cameral deposits are unusual in early cephalopods, but deposits similar to those identified here have been reported from the Ordovician *Bactroceras*⁶⁴. We interpret the deposits as a species related feature. Despite the abundance of calcareous shells in our thin section material, none of these show manganese

enrichments. This indicates that diagenetic features, such as a bulk rock-related enrichment, with manganese intruded by fluids or similar processes, can be excluded, because then it would also be present in other fossil shells. Also, there is no evidence for a diagenetic alteration affecting only the cephalopod specimens previous to their potential transport. Therefore, we interpret the presence of manganese to be a feature specific to the cephalopod taxon documented here, indicating that NFM F-2774 and F-2776 are conspecific.”

The siphuncle of all cephalopods consists of the septal necks and the connecting ring. Unless either of these structures are clearly shown, a cephalopod affinity or identity must remain doubtful, because it is the siphuncle - a very specific and unique structure within molluscs - that makes a cephalopod, and not aragonitic septa.

Unfortunately, the material of *Avaloniaceras* shows no such evidence. On the contrary, as far as I can judge, there is no evidence of a connecting ring (siphuncular wall in the authors terms) in the cross section of the first paratype in Fig. 1, but this should be visible with this type of preservation. Otherwise, if the connecting ring was destroyed during its early depositional history (either mechanical or by implosion), matrix would have been able to enter the chambers, or the calcite spar would have filled the entire space of the chamber, leaving no trace of the siphuncle in cross section except at the septum (i.e. septal neck). As discussed above, this is not the case here, because the septa are not visible. In the other cross-sectioned paratype in Fig. 2, there is a “siphuncular wall” visible, but this looks very similar to the outer shell wall and I do not think that this is the connecting ring. This issue could potentially be resolved by additional material where the possible siphuncle is exposed in longitudinal sections.

We agree with the Reviewer that both the connecting ring and the siphuncle are characteristic features of cephalopods. We have therefore provided new geochemical analysis and added a new paragraph to show that a connecting ring is present in our specimens. In order to account to Reviewer’s remarks on this, we’ve added lines 119–133 and 280–288 to the discussion. Our longitudinal section in Fig. 1 does not show such a structure, as the section does not cross the structure. In Fig. 3 two phosphoric rings are identified by element analysis (SEM-EDX maps) of thin sections within the phragmocone as an enrichment of P. We interpret these as remainders of the outer and inner walls of a connecting ring as it appears impossible to us to reproduce such a structure by cone-in-cone, telescoping or similar depositional or diagenetic effects. The separation between the inner and outer walls is relatively high and may be due to expansion during diagenesis. Virtually all early cephalopods documented in the literature have a calcitic composition of the connecting ring (e.g. Mutvei, 2002; Mutvei *et al.*, 2007; Mutvei, 2020). The only known exemption is the Ordovician *Bactroceras*, characterized by stable phosphatic connecting rings (Hewitt & Stait, 1985). According to the authors (Hewitt & Stait, 1985), the phosphatic connection ring allowed *Bactroceras* to populate deeper marine habitats. Connecting rings in our taxon were soft, and the scenario proposed for *Bactroceras* appears unlikely for our individuals which were found sediments indicating shallow habitats.

The phosphoric connecting ring was only identified by SEM-EDX mapping of thin sections. This indicates that preservation is highly dependent on the local burial history and it appears likely that the feature is not generally preserved and easily overlooked.

We agree with the Reviewer that the presence of a siphuncular and/or connecting ring is important for an unequivocal identification of cephalopods. Nevertheless, early cephalopods have repeatedly been documented from Laurentia without the explicit evidence of both features (e.g. Flower, 1964; Landing & Kröger, 2009). This material was cited in several recent publications as “undoubtful” or “unequivocal” cephalopods (e.g., Kröger & Landing, 2010; Kröger *et al.*, 2011; Kröger, 2013; Landing *et al.*, 2018; Fang *et al.*, 2019).

Fig. 3: SEM-EDX image showing the distribution of phosphorus in NFM F-2776. a, c) Phosphorus enrichment reveals the position of the inner (orange arrow) and outer wall (blue arrow) of the connecting ring. **b, d)** Detailed view of a and c. Scale bar is 1 mm.

The species appears to be very different from other (late) Cambrian cephalopods. While this is not an issue per se, it raises at least suspicions and opens up questions as to why there was

apparently such a drastic change in morphology. The most important differences are:
-Avaloniaceras has a relatively wide septal spacing. Late Cambrian cephalopods have extremely closely spaced chambers, usually about 0.05-0.1 in plectronocerids and below 0.2 in ellesmerocerids when compared to the corresponding shell diameter (Chen & Teichert 1983). You give a value of 0.6 for Avaloniaceras, but from the measurements and figures it becomes evident that this varies between 0.5 and 1.0, increasing during ontogeny (see also comment further below). The first cephalopods with similar (adult) chamber lengths are members of the Orthocerida, but they do not occur until the Ordovician. In my experience, relative chamber lengths above 0.3 are very rare before the Middle Ordovician. It has been argued that the close septal spacing helps counterbalancing buoyancy (e.g. Crick 1988), thus, Avaloniaceras would require alternative mechanisms that prevent it from floating to the surface. The other thing is that relative chamber length usually decreases during ontogeny. To my knowledge, there is no general study on this, but a similar trend has been shown in numerous taxa (e.g. Evans 2005; Kröger 2011; Aubrechtová 2015; personal observations in Cambrian taxa). The only exception is at the earliest ontogenetic stages, where there appears to be more variation (compare e.g. Kröger & Keupp 2004; Evans 2005).

We followed the suggestion of the Reviewer. We inserted a new paragraph on the septal spacing identified in our material and compared it with other late Cambrian cephalopods (lines 190–197):

“The septal spacing of late Cambrian cephalopods is usually small. The ratio of the septal spacing to the corresponding diameter of the shell is usually between 0.05–0.1 in plectronocerids and 0.2 in ellesmerocerids⁵⁹. Our material shows ratios of 0.5–0.6 and thus resembles Ordovician members of Orthocerida, although it should be noted that our measurements may be biased due to oblique sections. Alternatively, the wide septal space in the specimens documented here may correspond to an early ontogenetic stage. It has been shown that spaces between septa decrease during the ontogeny^{63,64}, although there is yet no evidence for this pattern in earliest members of cephalopods⁶⁵.”

Lastly, the authors explicitly cite the exponential growth of septal spacing as an argument for a cephalopod identity; however, in the cited article (Tajika et al. 2020) it is shown that chamber volume increases exponentially. Because of allometry, absolute septal spacing (length!) will increase more linearly, if volume increases exponentially. Therefore, septal spacing does not corroborate a cephalopod identity, but rather speaks against it. It may be argued that the specimens represent juveniles, but they would still be unusual in comparison with Cambrian cephalopods and I am unsure if much larger organisms could be expected in this assemblage.

Another problem is that Tajika et al. (2020) refer exclusively to coiled shells, while Avaloniaceras is straight.

We agree with the Reviewer and deleted this paragraph.

-Virtually all Cambrian cephalopods have a marginal siphuncle, which means that the siphuncle is touching the shell wall. Exceptions are very likely due to the section plane (note that this was not recognized by Chen & Teichert 1983; this argument is based partly on Wade & Stait 1993 and further unpublished personal observations). I have never seen any specimen or cross section of a Cambrian cephalopod where the siphuncle is demonstrably as far removed from the shell wall as in your specimen (i.e. more than its own diameter).

-The shape and position of the siphuncle cross section appears to vary extremely between the two cross-sectioned paratypes, based on Fig. 2 and your measurements in supplementary Fig. 6. The paratype in Fig. 2 has an extremely laterally compressed siphuncle cross section that is somewhat displaced compared to the symmetry axis. Usually, the siphuncle has a more or less circular cross section, but in that paratype the width/height ratio is apparently about 0.4, which is highly unusual and the only thing even remotely similar occurs in very specialised forms such as the piloceratids (the lowest I have seen is around 0.65).

-The species appears to be perfectly straight. Cambrian cephalopods are almost always at least slightly curved endogastrically or one side is more convex than the other, usually the dorsum. This may not be a significant difference, but it simply adds up.

We agree with the Reviewer that our material presents morphological features which differ from those of hitherto known cephalopods. We accept that measurements, even of the cross-sections, are potentially biased due to oblique cuts through our thin section material. This would partly explain the unusually high distance of the siphuncle to the wall, as well as the variation in the precise position. Compression of the siphuncle may have resulted from plastic diagenetic deformation. The length/width ratio of both cross-sections is 1.3. An oblique cut, could also explain the ellipsoid character. We agree that most early cephalopods are curved endogastrically, but straight forms are also known, e.g. *Dongshanoceras jiangshanense* Li, 1984 or *Ectenolites* Ulrich & Foerste, 1935 (Chen & Teichert, 1983; Li, 1984). We have added these points to our discussion (lines 134–137, 180–189 and 198–203).

You do not discuss alternative explanations. I have no expertise on SSFs, so I am in no good position to suggest any specific taxa, but perhaps this “siphuncle” could result from a small specimen washed into the body chamber of a larger specimen, much in the same way as in the holotype? The “siphuncle” in Fig. 2 looks quite similar to the surrounding small fossils. Note that there is a nice little paper on a similar phenomenon called “telescoping”, which may be useful to cite (Hladil et al. 2014).

We agree that our material might look similar to other fossils of the assemblage, but SEM-EDX revealed that its inner morphology differs substantially. In the present version of our manuscript we discuss the similarity with important taxa (including SSFs), such as *Tannuella* Missarzhevsky, 1969, *Salterella* Billings, 1861, *Knightoconus* Yochelson, Flower & Webers, 1973, *Volborthella* Schmidt, 1888, etc (e.g. lines 138–179). We also suggest that the superficial similarity might be the reason why our material was overlooked by earlier

researchers visiting this outcrop region. Please see above for the explanation on telescoping, cone-in-cone features and similar phenomena.

Another issue with the manuscript is that it ignores a large part of the existing literature dealing with Cambrian cephalopods and the only comparisons are made with *Plectronoceras*. While this genus is currently the oldest and you correctly state that it is poorly known (but arguably from much better material than *Avaloniaceras*!), only slightly younger cephalopods are known from rich assemblages in the latest Cambrian (Stage 10) of China, North America, Australia, Siberia and possibly Kazakhstan (Flower 1954, 1964; Chen et al. 1979a,b; Chen & Qi 1982; Chen & Teichert 1983; Li 1984; Wade 1988; Wade & Stait 1993; Mutvei et al. 2007; Fang et al. 2019; Dzik 2020). Chen & Teichert (1983) is probably the most important reference for Cambrian cephalopods and at least this should be cited somewhere. Amongst other things, it shows that starting with *Plectronoceras*, there is a continuous record of cephalopods until the latest Cambrian. While there has probably been a tendency to split and some of the species are probably based on oblique sections, it provides a good overview on the morphological range of Cambrian forms. At least some of these should be compared to the described material. Currently, most of the references cited by the authors are not dealing primarily with Cambrian cephalopods.

We followed the Reviewer's suggestion of citations and completed our references about Cambrian cephalopods and inserted a new paragraph in our revised manuscript (lines 180–189). We compared the morphology of these cephalopods, as also suggested by the Reviewer.

While you claim that the specimens are “well preserved” I would argue that the material hardly justifies the establishment of a new genus and species, at least not in its current state of preparation. The descriptions are based on sections that are likely oblique and off the symmetry plane, making it impossible to recognize crucial details of the siphuncle that would serve to distinguish it from any other early Palaeozoic cephalopod or even clearly identify it as one. The generic description in its current state would probably fit hundreds of species thus far described from the Cambro-Ordovician, while at the same time not mentioning the differences outlined above. The measurements are also doubtful and likely biased because the section is not in the symmetry plane. For example, the first chamber is measured from the apical end to the first internal septum. However, the apical end probably does not represent the last preserved septum, but rather where the section met the outer shell wall. Only if the symmetry plane is at least approximately known or at least the structure of the siphuncle would I consider that these specimens are worth the establishment of a new taxon, at least if cephalopod standards were to be applied. I therefore strongly advise against the description of a new species, at most *Ellesmerocerida* (?) indet. would be acceptable, if you insist on a cephalopod identity (it lacks any characters that would allow assigning it to the *Plectronocerida*). At the moment, I would also avoid the use of subclasses, because they may change in the future depending on the development of the new Treatise.

Even though we feel that the stratigraphic gap between our early Cambrian cephalopod and later Cambrian taxa would allow for the establishment of a new taxonomic unit, we follow the Reviewer's suggestion on referring to Ellesmerocerida(?) indet. We have re-measured specimen NFM-F-2774, as we identified remains of an additional septum (see also Reviewer's 2 remarks). Nevertheless, we are aware, that our measurements might be biased due to the potentially "oblique" cut of the section, as marked by Reviewers 1 and 2. We've added this point to our discussion (lines 192–194).

Title: I am not sure what you mean by "highly evolved"; the species presented here is in no way more "advanced" than other Cambrian or Early Ordovician cephalopods.

Our material is interpreted as a cephalopod of early Cambrian age, thus predating all other known cephalopods with a gap of c. 32 Ma. It might not be advanced as compared to later cephalopods, but more advanced than can be expected in the early Cambrian, given the current knowledge of cephalopod ancestors. However, we decided to follow the Reviewer's suggestion and changed the title.

Line 11: "c. 531 Ma": This seems a bit awkward, nowhere has a precise date for the appearance of cephalopods been proposed and if it was, it is pure speculation without any evidence. If you insist on leaving a date, I suggest you either round to "c. 530 Ma" or state a range within the early Cambrian. However, I recommend leaving out a date entirely, because this is essentially just a wild guess that cannot be falsified.

A precise date (c. 530 Ma) for the appearance of cephalopods has been proposed by Kröger *et al.* (2011). Nevertheless, we decided to follow Reviewer's suggestion and omitted the date.

Line 23-24: "Squids, cuttlefish, and octopuses originally possessed a chambered shell as the pearly Nautilus impressively demonstrates": This sentence is not entirely correct: 1) Sepia and Spirula are examples of recent coleoids that still possess a chambered shells, but in contrast to Nautilus, they are internal. 2) This may seem a bit pedantic, but it were not the squids, cuttlefish and octopuses themselves that originally possessed an (external) chambered shell, but their ancestors!

We followed the suggestion of the Reviewer and inserted "ancestors" (see also Reviewer's 2 comment).

Line 25: "probably benthic precursors": Citing Yochelson, Flower & Webers (1973) or Webers & Yochelson (1989) would be more appropriate here, even if you don't agree with their Knightoconus hypothesis. Alternatively, you may cite Kröger, Vinther & Fuchs (2011). We agree and inserted Kröger *et al.* (2011) as suggested by the Reviewer.

Line 31-33: "In members of the related order Ellesmerocerida, the shell is divided into chambers by concave-shaped septa, and the siphuncle is relatively wide and located

marginally along the ventral side of the phragmocone”: This is also the case in the Electronocera – in fact, apart from the wide ventral siphuncle, all characters that you mention are also found in a majority of Palaeozoic cephalopods.

Please see our answer on the taxonomic assignment above.

Line 51: “Shell breviconic”: the term “breviconic” is usually applied to species with high expansion rate, I would therefore consider this specimen as “longiconic” (alternatively, you could state the expansion rate/angle directly, because this is less subjective). Furthermore, both brevi- and longiconic species can be slightly curved (e.g. Electronoceras could also be considered as breviconic). Therefore, it is important to state the curvature, i.e. in this case either “straight” or “orthoconic”.

We follow the Reviewer’s suggestion and changed the terminology to “longiconic” and “straight” where applicable.

Line 81: “cone angle of 14-24°”: Keep in mind that the angle of expansion (this would be the correct term) is highly dependent on the section plane. If the section is oblique, the angle would be highly biased. The rounded anterior and posterior ends imply that the section is indeed oblique or off the symmetry plane (a way to experiment with this is by drawing a cone in a 3D drawing or geometry program and manually producing sections in different orientations and comparing their shapes).

AND

Line 91: “implicating a saggital rather than a longitudinal section”: As above, the section is not necessarily parallel to the median plane, but can also be oblique, thus biasing measurements. This is already a very serious problem for the large number of Chinese Cambrian taxa, which are almost exclusively known from thin sections. If further specimens are found, it will be difficult to tell whether any variation is intraspecific, due to species-differences or simply due to differing orientations of the section planes.

We agree with the Reviewer, as our measurements are biased. Thus, we deleted the measurement of the cone angle and added the possibility of a bias to the discussion (lines 192–194).

Line 92: “It is possible that the apertural and apical ends of the shell are missing in the specimen.”: Or probably outside the section plane.

We followed the suggestion of the Reviewer and inserted “or are outside the specimen”.

Line 99: “4.6 mm long and 3.6 mm wide”: When using directions in a cephalopod, any measurement in dorsoventral direction is “high”, in lateral direction “width” and in longitudinal direction “length”. Accordingly, it would be “high” instead of “long” here. Please check the manuscript carefully for similar mistakes, this can be very confusing to read.

We followed the suggestion of the Reviewer and checked this throughout the entire manuscript.

Line 152-174: Diagenesis: I am not familiar with SEM-EDX, so I cannot judge this part. However, it might be useful to cite similar studies on cephalopod or other originally aragonitic shells, if possible.

SEM-EDX is a well-known method all over the geologic disciplines. However, to our knowledge the technique has previously not been used in research on early cephalopods. Nevertheless, we found an example for hyoliths (Moysiuk *et al.*, 2017). We are convinced that SEM-EDX analysis is an important method to identify internal morphological features opaqued by diagenetic processes.

Line 176-184: Septal distance: I already explained my critique to this point above. I do not understand why you first state that chamber volumes can easily be estimated but then use only chamber length anyway. I wonder what would come out if you did the same analysis on another septate Cambrian mollusc. Would you expect this to differ from exponential growth? Also, I do not think that only four data points (out of which the first is questionable because it probably represents the shell margin rather than a septum) are statistically sufficient to support your conclusion.

We agree with the Reviewer and deleted the paragraph.

Line 202-214: This paragraph is held quite short, although the implications are quite far-reaching. It would mean that cephalopods evolved much earlier than expected (Plectronoceras is known from the late Jiangshanian, so the difference is probably closer to 30 million years instead of 20 as you suggest). Cephalopods have always been thought to undergo an explosive radiation in the latest Cambrian because they were able to colonize the free water column, thus occupying new niches. Your find would refute this hypothesis, because they would have been around for 30 million years in which they were apparently extremely rare. This leads to the question why it took them so long and what triggered the radiation in the latest Cambrian. The discussion on cephalopod ancestors is held quite short and no arguments are presented in relation to the evolution of the siphuncle, which is probably the most important open question in the search of the cephalopod ancestor. The research history of this subject is quite complex, see Webers & Yochelson (1989) for a detailed discussion of various hypotheses. Mutvei (2020) recently suggested yet another hypothesis which is highly speculative, but he assumes that simple siphuncular structures are inherently ancestral compared to complex ones, which I think is not necessarily the case. Another aspect that needs to be discussed in this paragraph is that according to Kröger (2013) and Fang *et al.* (2019), late Cambrian cephalopods are exclusively restricted to tropical palaeolatitudes. I am not familiar with the palaeogeography of Newfoundland, but if it fits also into this tropical scenario it would definitely be worth mentioning. Otherwise, this would suggest a high latitude origin of cephalopods with

subsequent migration and restriction to tropical regions. I do not know how likely this scenario would be.

We agree with the Reviewer and expanded our discussion on this topic significantly, based on his and Reviewer's 3 recommendations (lines 256–320). In particular, we discuss the open question of cephalopod ancestry (e.g., monoplacophoran, orthothecids, coleoid-like). We have added an enhanced discussion on the *Nectocaris* controversy as also recommended by Reviewer 3. Furthermore, we discuss the development of the siphuncle system based on the recent research of e.g. Mutvei (2020). Possible reasons for the gap in the fossil record are discussed. Lastly, as Reviewer 1 recommended, we have added a brief discussion on the paleobiogeography of early cephalopods and the situation of Avalonia, as well as the endemism of our material.

Figs.: The orientation of the cross sections is upside-down in all figures. The side with the siphuncle is generally regarded as the ventral side, therefore, this is supposed to be at the bottom.

We agree and changed the figure orientations.

Supplementary Fig. 2: The family range charts on the right side are directly taken from Fang et al. (2019) and should be cited!

We agree and added the reference of Fang *et al.* (2019).

Reviewer 2 (Martina Aubrechtová)

On line 23, I would suggest a small addition to increase accuracy of the statement: The ANCESTORS of squids, cuttlefish, and octopuses...

We follow the suggestion of the Reviewer and inserted “ancestors”, as also recommended by Reviewer 1.

Line 26 – please add the reference number 1 (Kröger et al. 2011) after “precursors”.

We agree and added the reference suggestion (please see also Reviewer 1).

Lines 28 – 33 – I strongly recommend to add some more citations. It is especially not sufficient to cite only the Treatise and a single other work here. The publications to consider particularly are:

KRÖGER, B., SERVAIS, T. & ZHANG, Y. 2009. The Origin and Initial Rise of Pelagic Cephalopods in the Ordovician. *PloS ONE* 4, e7262.

KRÖGER, B. & ZHANG, Y. 2009. Pulsed cephalopod diversification during the Ordovician. *Palaeogeography, Palaeoclimatology, Palaeoecology* 273, 174–183.

KRÖGER, B. 2013. Cambrian–Ordovician cephalopod palaeogeography and diversity, 429–448. In HARPER, D.A.T. & SERVAIS, T. (eds) *Early Palaeozoic*

Biogeography and Palaeogeography. Geological Society, London, Memoirs 38.
FANG, X., KRÖGER, B., ZHANG, Y.D., ZHANG, Y.B. and CHEN, T.E., 2019.
Palaeogeographic distribution and diversity of cephalopods during the Cambrian–Ordovician transition. *Palaeoworld*, 28(1-2), pp.51-57.

We agree and inserted all suggested citations at their respective positions of the manuscript.

In the Introduction, a similarity between Avaloniaceras and the Ellesmeroceratida is implied. However, the new genus is assigned to the Plectronoceratia with uncertain classification on the ordinal level. This may be confusing to the reader. Please, make clearer to what group you assign your new genus and why.

We agree with the Reviewer's comment. We refrain from the establishment of a new taxon and follow the Reviewer's 1 suggestion on referring to Ellesmerocerida(?) indet.

I have some concerns regarding the morphological descriptions of the specimens arising from the fact that the holotype is obliquely cut, out of the plane of symmetry. Under such circumstances, it is not possible to reliably measure the apical angle and it is also difficult to determine the shell shape. Besides that, when the shell is compressed in cross-section, there is not a single value of the apical angle but two because the shell expands more rapidly dorso-ventrally and less rapidly laterally. Please, kindly consider rephrasing.

We agree that the measurements are biased and deleted the cone-angle measurement. We've added the possible bias to our discussion. Please see also our response to Reviewer 1.

Also, I recommend a slightly more careful interpretation regarding increasing the chamber length along the shell. I am especially skeptical about whether the holotype really has such a long last phragmocone chamber. I would recommend double-checking whether there cannot be some septa secondarily missing.

We agree with the Reviewer's comment and double-checked our specimen. We have identified remains of other septa within the "last chamber" which are now added to our measurements and description, but we also spelled out that our measurements are potentially biased by oblique cuts through the phragmocone.

The photograph at Fig. 1a shows three rounded objects on the left side of the shell – can they be secondary deposit overgrowing remains of broken septa?

This is an intriguing comment. As this is a possibility, we have added it to our discussion. This would also fit the pattern of closely spaced septa commonly found in early cephalopods (see also Reviewer 1). However, in our opinion, there is not enough evidence for taking these possible remains into account for the biased measurements.

Why do you think the siphuncle was located ventrally?

We deleted the explicit interpretation of a ventrally located siphuncle throughout the manuscript.

Lines 99 + 105 – I suggest to express the dorso-ventral and lateral shell diameters also as relative values (i.e. compression rate).

We agree with the Reviewer's suggestion and added this to our description. Both specimens show a ratio of c. 1.3.

Please, include some more stratigraphic works, such as:

FLETCHER, T. P., 1972, Geology and Lower to Middle Cambrian trilobite faunas of southeast Avalon, Newfoundland, Ph.D. thesis, University of Cambridge, Cambridge.

BENUS, A. P., and LANDING, E. 1984, Depositional environments and biofacies of the Bonavista Formation (Early Cambrian, eastern Newfoundland), Geol. Soc. Am. Abstr. Progr. 16:3.

LANDING E. (1992) Lower Cambrian of Southeastern Newfoundland. In: Lipps J.H., Signor P.W. (eds) Origin and Early Evolution of the Metazoa. Topics in Geobiology, vol 10. Springer, Boston, MA

O'BRIEN, S.J. and KING, A.F., 2002. Neoproterozoic stratigraphy of the Bonavista Peninsula: preliminary results, regional correlations and implications for sediment-hosted stratiform copper exploration in the Newfoundland Avalon Zone. Current Research. Newfoundland Department of Mines and Energy, Geological Survey, Report, pp.02-1.

We have added Fletcher (1972) and Landing (1992) to our citations. Fletcher (2006), which was already cited, is the peer-reviewed publication based on his dissertation (Fletcher, 1972). Benus & Landing (1984) is a conference abstract. Benus (1988), which is already cited, is the respective peer-reviewed publication, providing more information. O'Brien & King (2002) focusses on the Bonavista Peninsula, a region which is approximately 60 km west of the study area and completely different to the latter one. Stratigraphy of the whole Avalon-Zone can be confusing, as nomenclature of localities, formations and other stratigraphic names is sometimes an unsolvable mixture due to its history.

Reviewer 3 (anonymous)

One of my main concerns are that the background literature referred to is sometimes slightly outdated. There exist a number of papers on Cambrian cephalopods that should be referenced in the introduction and discussion.

We agree, we inserted a number of new papers in the introduction and discussion, as also suggested by Reviewers 1 and 2.

Another, and more important issue is the fossil material at hand. The illustrated specimens are all from thin sections of bulk rock samples and only a single section of each specimen is

available. I realize that the nature of the material may make this difficult to accomplish, but more information would make the story much more viable. Have you tried to get 3D images through CT scanning of the blocks? It may not work if the density contrast is too low, but it may be worth trying. Alternatively, 3D models could be constructed through serial sectioning of some of the specimens. Compare the excellent results derived from the Silurian Herefordshire biota where the individual fossils are of similar size. I know this is a destructive technology, but it should work, and if you could show in a 3D model that the siphuncles cross multiple septa, no one would be able to doubt your interpretations. As it is now, you rely heavily on inferences from your interpretation of the diagenetic history, which will not convince everyone.

We agree, that a 3D model would provide many more information and would be of high interest for this paper. In preparation for the manuscript we have used grinding tomography (e.g. Pascual-Cebrian *et al.*, 2013a, b) and obtained 751 layers with a constant distance of 0.25 mm. Unfortunately, the composition of a suitable 3D model of a cephalopod failed, possibly due to improper layer distances and the deformation of the material (see Reviewer 1). From our experience, cephalopods are extremely rare and hard to identify in our material, as they are easily confused with SSF material (see Reviewer 1 and 3), making the chance to find individuals in a grinded section is low. Also, some structures might only be visible in SEM-EDX, or are even completely obscured by diagenesis. SEM-EDX would therefore be needed for each individual layer, making this unaffordable time-consuming. Unfortunately, micro-CT methodologies could not be used here as the lithological difference between sediment and shells is insufficient to provide usable images (see SEM-EDX maps).

I am also interested to learn more about your interpretations of the structures of the septa and siphuncles, particularly in comparison to what is known on this from the Cambrian Ellesmocerids and Plectronocerids. This would be interesting for any evolutionary discussion. We agree with the Reviewer's suggestion and added these points to our discussion (please see Reviewer 1).

In the comparison with other Cambrian shells, I see little reason to include *Coleoides*. It is very clearly very different. However, you should instead expand the comparison to *Allathea degeeri* to include orthothecid hyoliths more broadly. Many such taxa are septate and otherwise similar to cephalopod shells. In addition, some taxa are known to have preserved soft parts, so you have a base for more detailed discussion here, not least since this has been a hot research topic in the last 3-4 years. Also note that despite the fact that these fossils have recently been referred to the Lophophorates, other authors have strongly countered this interpretation and proposed a molluscan affinity for hyoliths.

We followed the Reviewer's suggestion and deleted the paragraph about *Coleoides* (original manuscript lines 146–150). We've added an enhanced discussion on orthothecids and the open question on the cephalopod ancestry (lines 256–320).

You mention the *Nectocaris* controversy, but are quick to discount it by a simple reference. I think the issue would benefit from a longer treatment in a paper like yours and I would recommend a summary of the arguments and your own evaluation of the evidence.

We agree, as the *Nectocaris* controversy is an intriguing topic and we have now added an enhanced discussion (lines 270–279).

And finally you end the paper by pointing out the 20 million year gap between your fossils and the first well established cephalopods. Maybe you can discuss the possible reasons or implications for this situation? It will undoubtedly be discussed in the future. Not least since the late Cambrian cephalopod faunas are so extremely diverse. Perhaps you can also compare to the fossil record of the other molluscan Classes and perhaps the hyoliths again? At least the bivalves have a somewhat similar Cambrian history.

We agree with the Reviewer and added the main points to our discussion (lines 138–179, see also Reviewer 1).

I have added some specific notes in the enclosed pdf file.

Line 30: We agree and added Kröger (2013) and Mutvei (2020).

Line 31: We agree and added Fang *et al.* (2019).

Line 32–33: We agree and added Mutvei *et al.* (2007).

Line 74–75: We agree and added the ornamented orthoconic fossils to the discussion.

Line 114: We agree and thus added a new figure (Fig 3) illustrating the connecting ring preserved as phosphorous remains, visible in SEM-EDX. Conspecificity of this specimen (NFM-F-2776) with NFM-F-2774 is evident by Mn-deposits in both specimens (see Reviewer 1 and discussion lines 214–224). The septa are prominent in specimen NFM-F-2774.

Line 124: We agree and added Flower (1954).

Line 146–148: Please see response on *Coleoloides* and *Allatheca degeeri*.

Line 205: Changed to Reviewer's suggestion and inserted Mutvei (2020).

Line 206: We agree and enhanced our discussion.

Line 218–220: Please see our comment on the *Nectocaris* controversy above.

Line 227–228: We agree and enhanced our discussion.

REFERENCES

Benus, A. Sedimentological context of a deep-water Ediacaran fauna (Mistaken Point Formatikon, Avalon Zone, eastern Newfoundland). In Trace Fossils, Small Shelly Fossils, and the Precambrian–Cambrian Boundary (eds Landing, E., Narbonne, G. M. & Myrow, P.). *New York State Museum and Geological Survey Bulletin* **463**, 8–9 (1988).

Chen, J.-Y. & Teichert, C. Cambrian Cephalopoda of China. *Palaeontographica*, Abteilung A **181**, 1–102 (1983).

Fang, X., Kröger, B., Zhang Y.-D., Zhang, Y.-B. & Chen, T.-E. Palaeogeographic distribution and diversity of cephalopods during the Cambrian–Ordovician transition. *Palaeoworld* **28**, 51–57, doi:<https://doi.org/10.1016/j.palwor.2018.08.007> (2019).

Fletcher, T. P. Geology and Lower to Middle Cambrian Trilobite Faunas of the Southwest Avalon, Newfoundland. *Unpublished PhD thesis, University of Cambridge*, 1–558 (1972).

Fletcher, T. P. Bedrock geology of the Cape St. Mary's Peninsula, Southwest Avalon Peninsula, Newfoundland (includes parts of NTS map sheets 1M/1, 1N/4, 1L/16 and 1K13). *Government of Newfoundland and Labrador, Geological Survey, Department of Natural Resources, St. John's, Report*, **06-02**, 1–117 (2006).

Flower, R. H. The nautiloid order Ellesmeroceratida (Cephalopoda). *New Mexico Bureau of Mines and Mineral Resources, Memoir* **12**, 1–164 (1964).

Flower, R.H. Cambrian cephalopods. *New Mexico Bureau of Mines and Mineral Resources, Bulletin* **40**, 1–51 (1954).

Hewitt, R. A. & Stait, B. Phosphatic connecting rings and ecology of an Ordovician ellesmerocerid nautiloid. *Alcheringa: An Australasian Journal of Palaeontology* **9**(3), 229–243, doi:[10.1080/03115518508618970](https://doi.org/10.1080/03115518508618970) (1985).

Kröger, B. Chapter 27 Cambrian–Ordovician cephalopod palaeogeography and diversity. In *Early Palaeozoic Biogeography and Palaeogeography* (eds Harper D. A. T. & Servais, T.). *Geological Society of London, Memoirs* **38**, 429–448, doi: <http://dx.doi.org/10.1144/M38.27> (2013).

Kröger, B. & Landing, E. Early Cambrian community evolution with eustatic change through the middle Beekmantown Group, northeast Laurentia. *Palaeogeography, Palaeoclimatology, Palaeoecology* **294**, 174–188 (2010).

Kröger, B., Vinther, J. & Fuchs, D. Cephalopod origin and evolution: A congruent picture emerging from fossils, development and molecules. *Bioessays* **33**, 602–613, doi:<https://doi.org/10.1002/bies.201100001> (2011).

Landing, E. Lower Cambrian of Southeastern Newfoundland. In *Origin and early Evolution of the Metazoa* (eds Lipps, J. H. & Signor, P. W.) 283–309 (Boston, MA, Springer US 1992).

Landing, E. & Kröger, B. The oldest cephalopods from east Laurentia. *Journal of Paleontology* **83**(1), 123–127, doi: <http://dx.doi.org/10.1666/08-078R.1> (2009).

Landing, E. *et al.* Early evolution of colonial animals (Ediacaran Evolutionary Radiation–Cambrian Evolutionary Radiation–Great Ordovician Biodiversification Interval). *Earth-Science Reviews* **178**, 105–135, doi:<https://doi.org/10.1016/j.earscirev.2018.01.013> (2018).

Li, L. Cephalopods from the Upper Cambrian Siyangshan Formation of western Zhejiang. In *Stratigraphy and Palaeontology of Systematic Boundaries in China, Cambrian–Ordovician Boundary* (1), 187–240 (Hefei: Anhui Science and Technology Publishing House 1984).

Moysiuk, J., Smith, M. R. & Caron, J.-B. Hyoliths are Palaeozoic lophophorates. *Nature* 541, 394–397, doi:<https://doi.org/10.1038/nature20804> (2017).

Mutvei, H. Connecting ring structure and its significance for classification of the orthoceratid cephalopods. *Acta Palaeontologica Polonica* **47**(1), 157–168 (2002).

Mutvei, H. Restudy of some plectronocerid nautiloids (Cephalopoda) from the late Cambrian of China; discussion on nautiloid evolution and origin of the siphuncle. *GFF*, doi: 10.1080/11035897.2020.1739742 (2020).

Mutvei, H., Zhang, Y.-B. & Dunca, E. Late Cambrian plectronocerid nautiloids and their role in cephalopod evolution. *Palaeontology* **50**(6), 1327–1333, doi: 10.1111/j.1475-4983.2007.00708.x (2007).

Pascual-Cebrian, E., Hennhöfer, D. K. & Götz, S. 3D morphometry of polyconitid rudist bivalves based on grinding tomography. *Facies* **59**, 347–358, doi:10.1007/s10347-012-0310-8 (2013a).

Pascual-Cebrian, E., Hennhöfer, D. & Goetz, S. High resolution and true colour grinding tomography of rudist bivalves, exemplified with the taxonomic revision of *Mathesia darderi* (Astre). *Caribbean Journal of Earth Science* **45**, 35–46 (2013b).

Reviewers' comments:

Reviewer #1 (Remarks to the Author):

Dear Authors and editor,

I reviewed the revised manuscript, and I was happy to see that there was a big effort to improve the manuscript and detailed responses to most of my criticized points, which I really appreciate. Amongst others, the cited literature is now much more appropriate and there is an expanded discussion on the early evolution of cephalopods. Despite this, I regret that I have to say that I am still not convinced that the specimens actually represent a cephalopod. It may be one, but I think that there is just not enough evidence to be sure, and personally, I am sceptical. I cannot point to any direction what it is instead, but I don't think that excluding other possibilities can be seen as definite evidence for a cephalopod identity, rather there should be unquestionable evidence that confirms it without requiring major interpretations about its diagenetic history. I strongly recommend that you make it clear that the cephalopod interpretation has its weaknesses and future studies with better material will need to confirm this hypothesis. You may still favour your hypothesis, but it would be good to more seriously consider alternative explanations, e.g. what if the specimens are not conspecific at all (i.e. NFM F-2774 has no siphuncle and NFM F-2776 has no septa, see below), how would you classify them? At the moment you are mostly listing what speaks for a cephalopod, and most evidence to the contrary is often dismissed in a short sentence (e.g. possible diagenetic origin of manganese enrichment, taphonomic effects, etc.) or just mentioned as unusual for a Cambrian cephalopod. Since a cephalopod identity would have such enormous phylogenetic, stratigraphic, palaeogeographic and morphological implications, it would also help people unfamiliar with early cephalopods to have different arguments listed. If it is published in its current state, I expect that it will be criticized in future studies and *Plectronoceras* will still be considered as the oldest undoubted cephalopod. Please find below some more details about my concerns.

In the revised version of the manuscript, you explain why the cross sections and the longitudinal sections should be assigned to the same taxon, i.e. by internal manganese enrichment, which is absent in all other fossils of the assemblage. You interpret the manganese-enriched structures as cameral deposits. However, to me they are not reminiscent of any biologically precipitated structures that were originally part of the organism (perhaps excluding microbes that may have grown inside the chambers?), since their organisation appears to be arbitrary, in contrast to cameral deposits in cephalopods (compare e.g. Fischer & Teichert 1969; Blind 1991; Seuss et al. 2012; Pohle & Klug 2018; note that Mutvei 2018 still considers cameral deposits to be post-mortem, but this opinion is not followed by most workers). They also seem to be relatively depleted in calcium (figs. 1f, 2e), which is contrasting to cephalopod cameral deposits, which are thought to be aragonitic or calcitic, though studies on this are rather rare (Seuss et al. 2012). The cameral deposits in *Bactroceras latisiphonatum* (synonymized with *B. angustisiphonatum*) as reported by Hewitt & Stait (1985) have been questioned by Evans (2005). Aubrechtová (2015) also stated that cameral deposits are absent in *Bactroceras*. Cameral deposits seem to be restricted to the Orthoceratoidea (or Orthoceratia) and would thus not be expected in Cambrian Ellesmerocerids (King & Evans 2019). Note that *Bactroceras* has been removed from the Ellesmerocerida and is currently classified within the Orthocerida or Riocerida (Aubrechtová 2015; King & Evans 2019). This means that if cameral deposits in the specimens at hand were accepted, it would either create an even larger stratigraphic gap (Cambrian cephalopods do not contain cameral deposits) or cameral deposits evolved independently later. In conclusion, it is almost certain that there are no cameral deposits (at least homologous to those in Ordovician taxa) in these specimens, for morphological, chemical, stratigraphic and phylogenetic reasons. The question is then, where does the manganese enrichment come from? I do not know if there is any precedence for biomineralized structures with high manganese content in multicellular organisms. Furthermore, although I am not familiar with diagenetic processes that could lead to manganese enrichment, I am not sure if they can be excluded solely on the basis that they are rare and occur only in these fossils. Note that there is some danger of circular reasoning here: "Manganese occurs only in these specimens, so they must be conspecific. Because no other species at the locality contains manganese, it cannot be diagenetic in origin." (I am exaggerating a bit, but I hope you see my point). As far as I understand, the manganese enrichment is the only thing linking these

two types of specimens. Therefore, you need to be absolutely sure that it is not diagenetic!

Additionally, new analyses were supplied to present evidence for a connecting ring (fig. 3). In your interpretation, the phosphor enrichment presents evidence for a connecting ring. However, the phosphor enrichment is only visible as a very thin discontinuous layer along the inner and outer surfaces of the alleged connecting ring. Furthermore, it is described as a phosphatic connecting ring, even though the structure itself is practically devoid of phosphor, except for the inner and outer surfaces. These surfaces are also termed as "inner" and "outer wall" of the connecting ring, but this does not correspond to a cephalopod, rather it usually consists of an inner and an outer layer (the latter may be destroyed during diagenesis), which are homogenous by themselves (see e.g. Mutvei 2002). If I am interpreting the SEM-EDX images correctly, the "connecting ring" seems to consist mostly of calcium (fig. 2e) and may be indistinguishable from the shells of other SSF in the assemblage. One may just as well interpret the small lining close to the right margin of the specimen in fig. 3a as inner surface of the outer shell wall. Considering that other SSFs outside of the specimen in fig. 3 seem to be depleted in phosphor as well, I am not sure whether there is a large difference in composition of shell material when compared to the "connecting ring". Also, the amount of phosphor seems rather minor, especially considering that similar amounts occur in the chamber, and the matrix is apparently full of phosphor. Is this really a significant amount or could it be that it just accumulated on the surface of the structure (I am speculating, but perhaps small mud particles were dissolved in the water contained in the chamber and just sunk down)? In summary, I think that there is no clear evidence that confirms this structure as a connecting ring and to me it also does not differ significantly from normal shell material.

Following from the above, I still think that reproducing such a structure by telescoping or cone-in-cone would be possible, especially as this phenomenon does seem to occur relatively frequently in the layer (fig. 1, some cases maybe also in fig. 4). The main difference between these and the cross sections containing a "siphuncle" is that the rest of the specimen is filled with calcite spar, but is it not possible that it was sealed early during diagenesis, but a smaller specimen (already filled with matrix) was washed inside before the sealing? Even if this is unlikely, the "cephalopods" seem to be rare after all. If you think that it is impossible, then it should be stated more clearly why, at the moment you just say "It is impossible to reproduce such a structure only with cone-in-cone, telescoping, or similar depositional or diagenetic effects."

You state that unequivocal cephalopods have been identified on other occasions without necessarily showing septal necks and connecting ring (e.g. Flower 1964; Landing & Kröger 2009). However, I would argue that these two cases are a bit less problematic because roughly coeval cephalopods were known from the same region and they also more closely resemble known Cambrian cephalopods, e.g. the cephalopods described by Landing & Kröger (2009) have very close cameral spacing (below 1 mm) and are cyrtoconic, so if a siphuncle was visible, it would be almost identical to *Plectronoceras*. The same can unfortunately not be said about the presented material. Flower (1964) regarded *Shelbyoceras* as a possible cephalopod, but also noted that the siphuncle was unknown – this genus was in fact shortly thereafter shown to represent a monoplacophoran (Stinchcomb & Echols 1966). Since the new material would push back the origin of cephalopods by 30 million years, I think that waterproof evidence for a cephalopod identity would be required, especially keeping in mind the long history of alleged early cephalopods which later turned out to be no cephalopods at all. In that sense, I am afraid that only a longitudinal section which clearly shows the siphuncle, or a 3D reconstruction can resolve the issue.

Another small remark to the 3D reconstruction, you responded to Reviewer 3 that you tried this but were unsuccessful. SEM-EDX on individual layers would be required to look for these specimens, which would be too time-consuming. Are there really no specimens like the one in fig. 1 or fig. 2? This would not require SEM-EDX and you should be able to at least see whether septa and siphuncle are present, or whether the "siphuncle" is just a taphonomic phenomenon. In case that there are none of these specimens to be seen in the entire image stack, are the rock samples from the illustrated samples still available? If yes, you could try to use serial grinding tomography on e.g. NFM F-2776 (fig. 2), ideally with a high resolution? Even if this results only in a partially reconstructed model with relatively low resolution it should at least clarify whether septa are present and the three-dimensional structure of the potential siphuncle.

Some minor remarks:

Line 125: Actually, similar connecting rings have been described by Mutvei (2017). However, note that some of the newer publications by Harry Mutvei are controversial and especially his new taxonomic units are not widely applied (King & Evans 2019).

Line 195-198: I think there is a slight misunderstanding of what I said before. From personal experience, I think that the decrease in cameral spacing during ontogeny is widespread in early Palaeozoic cephalopods, but there is no study that explicitly shows this as a general pattern. As for Cambrian taxa, I have personally observed this in Chinese and Australian specimens (unpublished). In general, this variation is very gradual and there are no large jumps in cameral spacing. The smallest specimens I have observed are about 2-3 mm at their broken apical end and have chambers below 1 mm. So while there is a decrease during ontogeny, this decreases from at maximum about 0.25 in the earliest observed stages; in plectronocerids, cameral spacing doesn't even go above 0.1. I would also leave the citation of Kröger & Keupp (2004) out, as it is not relevant in this context.

Line 199-204: I cannot image a way to obliquely cut a phragmocone with a marginal siphuncle (i.e. continuously touching the shell wall) that would remove the siphuncle from the shell wall ...

Line 227-230: I would call it an observation by Chen & Teichert rather than a hypothesis. But Mg enrichment seems to occur everywhere, even in the matrix, is this really good evidence for septa?

Line 294: Well, but it arguably opens up an even larger gap between this cephalopod and the next youngest ...

Figs. 1+2: The specimens are still mentioned as holotype and paratype in the figure captions.

I apologize if I sound overly critical, it is not my intention to discredit your work. You have obviously done a lot of work. I am just very sceptical about this and I hope you can understand my point of view.

Best wishes,
Alexander Pohle

References:

- Aubrechtová, M. 2015: A revision of the Ordovician cephalopod *Bactrites sandbergeri* Barrande: Systematic position and palaeobiogeography of *Bactroceras*. *Geobios* 48, 193–211.
- Blind, W. 1991: Über Anlage und Funktion von Kammerablagerungen in Orthoceren-Gehäusen. *Palaeontographica Abteilung A* 218, 35–47.
- Evans, D.H. 2005: The Lower and Middle Ordovician cephalopod faunas of England and Wales. *Monograph of the Palaeontographical Society* 158, 1–81.
- Fischer, A.G. & Teichert, C. 1969: Cameral deposits in cephalopod shells. *The University of Kansas Paleontological Contributions* 37, 1–30.
- Flower, R.H. 1964: The nautiloid order Ellesmeroceratida (Cephalopoda). *New Mexico Bureau of Mining & Mineral Resources, Memoir* 12, 1–164.
- Hewitt, R.A. & Stait, B. 1985: Phosphatic connecting rings and ecology of an ordovician ellesmerocerid nautiloid. *Alcheringa*.
- King, A.H. & Evans, D.H. 2019: High-level classification of the nautiloid cephalopods: a proposal for the revision of the Treatise Part K. *Swiss Journal of Palaeontology* 138, 65–85.
- Landing, E. & Kröger, B. 2009: The oldest cephalopods from east Laurentia. *Journal of Paleontology* 83, 123–127.
- Mutvei, H. 2002: Connecting ring structure and its significance for classification of the orthoceratid cephalopods. *Acta Palaeontologica Polonica* 47, 157–168.
- Mutvei, H. 2017: The new order Mixosiphonata (Cephalopoda: Nautiloidea) and related taxa; estimations of habitat depth based on shell structure. *GFF* 139, 219–232.
- Mutvei, H. 2018: Cameral deposits in Paleozoic cephalopods. *GFF* 140, 254–263.

Pohle, A. & Klug, C. 2018: Early and Middle Devonian cephalopods from Hamar Laghdad (Tafilalt, Morocco) and remarks on epicoles and cameral deposits. *Neues Jahrbuch für Geologie und Paläontologie - Abhandlungen* 290, 203–240.

Seuss, B., Mapes, R.H., Klug, C. & Nützel, A. 2012: Exceptional Cameral Deposits in a Sublethally Injured Carboniferous Orthoconic Nautiloid from the Buckhorn Asphalt Lagerstätte in Oklahoma, USA. *Acta Palaeontologica Polonica* 57, 375–390.

Stinchcomb, B.L. & Echols, D.J. 1966: Missouri Upper Cambrian Monoplacophora previously considered cephalopods. *Journal of Paleontology* 40, 647–650.

Author's response to Reviewers' comments

Our manuscript COMMSBIO-20-1975A entitled "A primordial cephalopod from the early Cambrian of southeastern Newfoundland" has been reviewed again by Alexander Pohle to whom we like to express our gratitude. He perceived several weaknesses in the current form of our manuscript, to which we respond below. The comments are highlighted in blue, each answer is beneath.

Reviewer #1 (Remarks to the Author):

Dear Authors and editor,

I reviewed the revised manuscript, and I was happy to see that there was a big effort to improve the manuscript and detailed responses to most of my criticized points, which I really appreciate. Amongst others, the cited literature is now much more appropriate and there is an expanded discussion on the early evolution of cephalopods. Despite this, I regret that I have to say that I am still not convinced that the specimens actually represent a cephalopod. It may be one, but I think that there is just not enough evidence to be sure, and personally, I am sceptical. I cannot point to any direction what it is instead, but I don't think that excluding other possibilities can be seen as definite evidence for a cephalopod identity, rather there should be unquestionable evidence that confirms it without requiring major interpretations about its diagenetic history. I strongly recommend that you make it clear that the cephalopod interpretation has its weaknesses and future studies with better material will need to confirm this hypothesis. You may still favour your hypothesis, but it would be good to more seriously consider alternative explanations, e.g. what if the specimens are not conspecific at all (i.e. NFM F-2774 has no siphuncle and NFM F-2776 has no septa, see below), how would you classify them? At the moment you are mostly listing what speaks for a cephalopod, and most evidence to the contrary is often dismissed in a short sentence (e.g. possible diagenetic origin of manganese enrichment, taphonomic effects, etc.) or just mentioned as unusual for a Cambrian cephalopod. Since a cephalopod identity would have such enormous phylogenetic, stratigraphic, palaeogeographic and morphological implications, it would also help people unfamiliar with early cephalopods to have different arguments listed. If it is published in its current state, I expect that it will be criticized in future studies and *Plectronoceras* will still be considered as the oldest undoubted cephalopod. Please find below some more details about my concerns.

In the revised version of the manuscript, you explain why the cross sections and the longitudinal sections should be assigned to the same taxon, i.e. by internal manganese enrichment, which is absent in all other fossils of the assemblage. You interpret the manganese-enriched structures as cameral deposits. However, to me they are not reminiscent of any biologically precipitated structures that were originally part of the organism (perhaps excluding microbes that may have grown inside the chambers?), since their organisation appears to be arbitrary, in contrast to cameral deposits in cephalopods (compare e.g. Fischer & Teichert 1969; Blind 1991; Seuss et al. 2012; Pohle & Klug 2018; note that Mutvei 2018

still considers cameral deposits to be post-mortem, but this opinion is not followed by most workers). They also seem to be relatively depleted in calcium (figs. 1f, 2e), which is contrasting to cephalopod cameral deposits, which are thought to be aragonitic or calcitic, though studies on this are rather rare (Seuss et al. 2012). The cameral deposits in *Bactroceras latisiphonatum* (synonymized with *B. angustisiphonatum*) as reported by Hewitt & Stait (1985) have been questioned by Evans (2005). Aubrechtová (2015) also stated that cameral deposits are absent in *Bactroceras*. Cameral deposits seem to be restricted to the Orthoceratoidea (or Orthoceratia) and would thus not be expected in Cambrian Ellesmerocerids (King & Evans 2019). Note that *Bactroceras* has been removed from the Ellesmerocerida and is currently classified within the Orthocerida or Riocerida (Aubrechtová 2015; King & Evans 2019). This means that if cameral deposits in the specimens at hand were accepted, it would either create an even larger stratigraphic gap (Cambrian cephalopods do not contain cameral deposits) or cameral deposits evolved independently later. In conclusion, it is almost certain that there are no cameral deposits (at least homologous to those in Ordovician taxa) in these specimens, for morphological, chemical, stratigraphic and phylogenetic reasons.

The question is then, where does the manganese enrichment come from? I do not know if there is any precedence for biomineralized structures with high manganese content in multicellular organisms. Furthermore, although I am not familiar with diagenetic processes that could lead to manganese enrichment, I am not sure if they can be excluded solely on the basis that they are rare and occur only in these fossils. Note that there is some danger of circular reasoning here: “Manganese occurs only in these specimens, so they must be conspecific. Because no other species at the locality contains manganese, it cannot be diagenetic in origin.” (I am exaggerating a bit, but I hope you see my point). As far as I understand, the manganese enrichment is the only thing linking these two types of specimens. Therefore, you need to be absolutely sure that it is not diagenetic!

Additionally, new analyses were supplied to present evidence for a connecting ring (fig. 3). In your interpretation, the phosphor enrichment presents evidence for a connecting ring. However, the phosphor enrichment is only visible as a very thin discontinuous layer along the inner and outer surfaces of the alleged connecting ring. Furthermore, it is described as a phosphatic connecting ring, even though the structure itself is practically devoid of phosphor, except for the inner and outer surfaces. These surfaces are also termed as “inner” and “outer wall” of the connecting ring, but this does not correspond to a cephalopod, rather it usually consists of an inner and an outer layer (the latter may be destroyed during diagenesis), which are homogenous by themselves (see e.g. Mutvei 2002). If I am interpreting the SEM-EDX images correctly, the “connecting ring” seems to consist mostly of calcium (fig. 2e) and may be indistinguishable from the shells of other SSF in the assemblage. One may just as well interpret the small lining close to the right margin of the specimen in fig. 3a as inner surface of the outer shell wall. Considering that other SSFs outside of the specimen in fig. 3 seem to be depleted in phosphor as well, I am not sure whether there is a large difference in composition of shell material when compared to the “connecting ring”. Also, the amount of phosphor seems rather minor, especially considering that similar amounts occur in the chamber, and the matrix is apparently full of phosphor. Is this really a significant amount or could it be that it just accumulated on the surface of the structure (I am speculating, but

perhaps small mud particles were dissolved in the water contained in the chamber and just sunk down)? In summary, I think that there is no clear evidence that confirms this structure as a connecting ring and to me it also does not differ significantly from normal shell material.

Following from the above, I still think that reproducing such a structure by telescoping or cone-in-cone would be possible, especially as this phenomenon does seem to occur relatively frequently in the layer (fig. 1, some cases maybe also in fig. 4). The main difference between these and the cross sections containing a “siphuncle” is that the rest of the specimen is filled with calcite spar, but is it not possible that it was sealed early during diagenesis, but a smaller specimen (already filled with matrix) was washed inside before the sealing? Even if this is unlikely, the “cephalopods” seem to be rare after all. If you think that it is impossible, then it should be stated more clearly why, at the moment you just say "It is impossible to reproduce such a structure only with cone-in-cone, telescoping, or similar depositional or diagenetic effects."

You state that unequivocal cephalopods have been identified on other occasions without necessarily showing septal necks and connecting ring (e.g. Flower 1964; Landing & Kröger 2009). However, I would argue that these two cases are a bit less problematic because roughly coeval cephalopods were known from the same region and they also more closely resemble known Cambrian cephalopods, e.g. the cephalopods described by Landing & Kröger (2009) have very close cameral spacing (below 1 mm) and are cyrtconic, so if a siphuncle was visible, it would be almost identical to *Plectonoceras*. The same can unfortunately not be said about the presented material. Flower (1964) regarded *Shelbyoceras* as a possible cephalopod, but also noted that the siphuncle was unknown – this genus was in fact shortly thereafter shown to represent a monoplacophoran (Stinchcomb & Echols 1966). Since the new material would push back the origin of cephalopods by 30 million years, I think that waterproof evidence for a cephalopod identity would be required, especially keeping in mind the long history of alleged early cephalopods which later turned out to be no cephalopods at all. In that sense, I am afraid that only a longitudinal section which clearly shows the siphuncle, or a 3D reconstruction can resolve the issue.

Although we are convinced about the cephalopod identity of our material, we accept that there is no unequivocal evidence. Thus, we have changed the title and the entire manuscript accordingly. The reviewer, to whom we want to express our thanks for the extensive effort put into the constructive reviews, is particularly concerned regarding the existence of the phosphorous ring clearly visible in Fig. 3. As written in the figure captions of the SEM pictures: the brighter the colour, the higher the amount of the element. Even low amounts or traces of an element are revealed with this method. It is highly unlikely to find measurable concentrations of phosphorus arranged in a perfectly circular shape and not randomly distributed, while the rest of the specimen (excluding the shell parts) is almost devoid of phosphorus. Such a ring structure is not visible anywhere else within this specimen. We agree that the apparent connecting ring preserved here is unique and does not fit current knowledge of cephalopods. However, it appears that an oddly developed connecting ring or a different, yet unknown biological structure is a likely scenario, compared to a speculation

contemplating any dissolved and precipitated mud material. Although intriguing, the latter hypothesis is geologically (chemically, sedimentologically, etc.) impossible, even if the system was not sealed. Typical matrix elements (clay minerals!), such as K, Al and Si are absent in this part of the specimen. If a precipitation of clay material existed, these elements would have vanished randomly from the system, leaving only the phosphorus, which is even more unlikely. We appreciate the recommendation to clarify weaknesses in our interpretation and have therefore added a new subchapter in the discussion (lines 311–352), which discusses the evidence itself and the points of the reviewer:

“Unequivocal cephalopod identity?”

We are aware that an unequivocal cephalopod assignment of the specimens described and discussed here must await future findings of better-preserved material less affected by diagenesis. For example, no siphuncle was established in NFM F-2774 and uncertainties thus remain about the conspecificity of this specimen and others presented here (e.g. NFM F-2776). Where a siphuncle is identified, it is situated peripherally, but not as marginally as expected, and it may originally have consisted of soft phosphorous material. Our interpretation of cameral deposits is at odds with their absence in other Cambrian cephalopods. The biologic origin of these deposits is also uncertain as manganese is contained within the diagenetic cement; this condition differs from cameral deposits in other Palaeozoic cephalopods^{79–82}. Thus, the manganese-bearing deposits may not be cameral deposits *sensu stricto* but biological enrichments of uncertain origin or, although unlikely, are diagenetic features.

The phosphorous-bearing outer and calcitic inner walls of the potential connecting ring in NFM F-2776 (Figs 2, 3) are also remarkably unusual features in fossil cephalopods, as siphuncle and connecting ring structures should be homogeneous by themselves²⁹. We can therefore not exclude a scenario in which phosphorus enrichment resulted from cone-in-cone processes, in particular as SSFs presenting these structures are present in the limestone layer investigated here and are readily identified in cross sections of individuals of this assemblage. Nevertheless, the potential cephalopod described here, including NFM F-2776, is easily distinguished from these SSF elements by a spar envelope around the apparent siphuncle. Also, a scenario invoking cone-in-cone processes would be complex and include the following stages: (1) deposition of the original specimen, (2) infill of another conical fossil already filled with sediment, whereas the original specimen is still devoid of sediment, and finally (3) sealing of the entire composition during diagenesis. This scenario would still not explain the phosphoric ring identified in the interior of NFM F-2776 (Fig. 3). Therefore, this latter scenario requires an additional prerequisite, i.e. soft sediment rich in P, Ca and possibly Mg accumulating in the specimen and there precipitating the above elements in circular shape around the washed-in specimen, which was later destroyed by diagenesis. Typical matrix elements such as K, Al and Si are absent in this part of NFM F-2776. Clearly, this alternative scenario requires numerous assumptions. Applying Occam’s razor, we therefore favour the interpretation that the phosphorus enrichment is evidence for a connecting ring.

The present material clearly differs from typical early Cambrian septate molluscs, hyoliths and other SSFs, e.g. in septal morphology and shell structure, and was therefore not assigned to a known early Cambrian taxon. However, a failed assignment to a known taxon combined with uncertain cephalopod features, does not constitute unequivocal cephalopod evidence. We

therefore provisionally allocate a cephalopod identity of our specimens, pending better-preserved material, which would have extensive phylogenetic, stratigraphic, palaeogeographic and morphological implications. Based on the present data we suggest that future search for early and middle Cambrian cephalopods should focus on orthoconic fossils with close septal distances, a connecting ring and a siphuncle preserved as soft, phosphorous material.”

Another small remark to the 3D reconstruction, you responded to Reviewer 3 that you tried this but were unsuccessful. SEM-EDX on individual layers would be required to look for these specimens, which would be too time-consuming. Are there really no specimens like the one in fig. 1 or fig. 2? This would not require SEM-EDX and you should be able to at least see whether septa and siphuncle are present, or whether the “siphuncle” is just a taphonomic phenomenon. In case that there are none of these specimens to be seen in the entire image stack, are the rock samples from the illustrated samples still available? If yes, you could try to use serial grinding tomography on e.g. NFM F-2776 (fig. 2), ideally with a high resolution? Even if this results only in a partially reconstructed model with relatively low resolution it should at least clarify whether septa are present and the three-dimensional structure of the potential siphuncle.

We absolutely agree, that it would be nice to have such a reconstruction, but as explained in the first answer to Reviewer 3, we have tried this and failed. We hope that future studies will be able to solve this.

Some minor remarks:

Line 125: Actually, similar connecting rings have been described by Mutvei (2017). However, note that some of the newer publications by Harry Mutvei are controversial and especially his new taxonomic units are not widely applied (King & Evans 2019). A highly welcome hint. We have added Mutvei (2017) as a reference here.

Line 195-198: I think there is a slight misunderstanding of what I said before. From personal experience, I think that the decrease in cameral spacing during ontogeny is widespread in early Palaeozoic cephalopods, but there is no study that explicitly shows this as a general pattern. As for Cambrian taxa, I have personally observed this in Chinese and Australian specimens (unpublished). In general, this variation is very gradual and there are no large jumps in cameral spacing. The smallest specimens I have observed are about 2-3 mm at their broken apical end and have chambers below 1 mm. So while there is a decrease during ontogeny, this decreases from at maximum about 0.25 in the earliest observed stages; in plectronocerids, cameral spacing doesn't even go above 0.1. I would also leave the citation of Kröger & Keupp (2004) out, as it is not relevant in this context.

We appreciate the reviewer's clarification here, and we would certainly be happy to cite his observation, but as this data is not published, we have decided to rather delete this source of speculation as the particular point is not overly important for the main line of argumentation.

Line 199-204: I cannot image a way to obliquely cut a phragmocone with a marginal siphuncle (i.e. continuously touching the shell wall) that would remove the siphuncle from the

shell wall ...

We agree with the reviewer here and have deleted these lines.

Line 227-230: I would call it an observation by Chen & Teichert rather than a hypothesis. But Mg enrichment seems to occur everywhere, even in the matrix, is this really good evidence for septa?

We agree and changed “hypothesis” to “observation”. Mg enrichment is in the matrix due to the minerals contained in the matrix. Mg enrichment is not in the specimen elsewhere than in the shell, the septa, and regions with possible broken septa (Fig 1). Thus, Mg is in fact an almost perfect evidence for septa.

Line 294: Well, but it arguably opens up an even larger gap between this cephalopod and the next youngest ...

We agree, but line 294 refers to other high-level taxa, not other cephalopods, a completely different topic and scope. We have discussed the gap to other cephalopods throughout the sub-chapter.

Figs. 1+2: The specimens are still mentioned as holotype and paratype in the figure captions.

We agree and have changed both.